# Corner Gradient Descent

## Abstract

We consider SGD-type optimization on infinite-dimensional quadratic problems with power law spectral conditions. It is well-known that on such problems deterministic GD has loss convergence rates $L_t = O(t^{-\varsigma})$, which can be improved to $L_t = O(t^{-2\varsigma})$ by using Heavy Ball with a non-stationary Jacobi-based schedule (and the latter rate is optimal among fixed schedules). However, in the mini-batch Stochastic GD setting, the sampling noise causes the Jacobi HB to diverge; accordingly no $O(t^{-2\varsigma})$ algorithm is known. In this paper we show that rates up to $O(t^{-2\varsigma})$ can be achieved by a generalized stationary SGD with infinite memory. We start by identifying generalized (S)GD algorithms with contours in the complex plane. We then show that contours that have a corner with external angle $\theta\pi$ accelerate the plain GD rate $O(t^{-\varsigma})$ to $O(t^{-\theta\varsigma})$. For deterministic GD, increasing $\theta$ allows to achieve rates arbitrarily close to $O(t^{-2\varsigma})$. However, in Stochastic GD, increasing $\theta$ also amplifies the sampling noise, so in general $\theta$ needs to be optimized by balancing the acceleration and noise effects. We prove that the optimal rate is given by $\theta_{\max} = \min(2, \nu, \frac{2}{\varsigma + 1/\nu})$, where $\nu, \varsigma$ are the exponents appearing in the capacity and source spectral conditions. Furthermore, using fast rational approximations of the power functions, we show that ideal corner algorithms can be efficiently approximated by practical finite-memory algorithms.

## 1 Introduction

It is well-known that Gradient Descent (GD) on quadratic problems can be accelerated using the additional momentum term (the "Heavy Ball" algorithm, [19]). For ill-conditioned problem, Heavy Ball with a suitable non-stationary ("Jacobi") predefined schedule allows to accelerate a power-law loss converge rate $O(t^{-\varsigma})$ to $O(t^{-2\varsigma})$, i.e. double the exponent $\varsigma$ [8, 16]. This acceleration is the best possible for non-adaptive schedules.

On the other hand, for mini-batch *Stochastic* Gradient Descent (SGD) typically used in modern machine learning, the convergence rate picture is much more complicated, and much less is known about possible acceleration. The natural quadratic problem in this case is the fitting of a linear model with a sampled quadratic loss. In the power-law spectral setting, it was found in [4] that plain SGD has two distinct convergent phases: either the sampling noise is weak and the SGD rate is the same $O(t^{-\varsigma})$ as for GD, or the convergence is slower due to the prevalence of the sampling noise. We refer to these two scenarios as *signal-* and *noise-dominated*, respectively.

This picture was refined in several other works [18, 23, 24, 25, 29]. In particular, [29] examined generalized SGDs with finite linear memory of any size (generalizing the momentum and similar terms) and proved that with stationary schedules they all have the same phase diagram as plain SGD (Figure 2 left); in particular, they do not accelerate the plain GD/SGD rate $O(t^{-\varsigma})$.

On the other hand, the non-stationary Jacobi Heavy Ball accelerating deterministic GD from $O(t^{-\varsigma})$ to $O(t^{-2\varsigma})$ fails for mini-batch Stochastic GD: it eventually starts to diverge due to the accumulating

Submitted to 39th Conference on Neural Information Processing Systems (NeurIPS 2025). Do not distribute.

sampling noise. [23] have proposed a non-stationary modification of SGD that achieves a quadratic acceleration, but only on finite-dimensional problems. [29] have proposed a non-stationary modification of the Heavy Ball/momentum algorithm that is heuristically expected (but not yet proved) to achieve rates $O(t^{-\theta\varsigma})$ with some $1 < \theta < 2$ on infinite-dimensional problems.

To sum up, the topic of SGD acceleration in ill-conditioned quadratic problems is far from settled.

In the present paper we propose an entirely new approach to acceleration of (S)GD that both provides a new general geometric viewpoint and proves that, in a certain rigorous sense, SGD in the signal-dominated regime can be accelerated from $O(t^{-\varsigma})$ to $O(t^{-\theta\varsigma})$ with $\theta$ up to 2.

**Our contributions:**

1. **A view of generalized (S)GD as contours** (Section 3). We show that stationary (S)GD algorithms with an arbitrary-sized linear memory can be identified with contours in the complex plane. This identification leverages the characteristic polynomials $\chi$ and the loss expansions of memory-$M$ (S)GD from [29]. We show that all the information needed to compute the loss evolution is contained in a response map $\Psi : \{z \in \mathbb{C} : |z| \geq 1\} \to \mathbb{C}$ associated with $\chi$. The map $\Psi$ gives rise to the contour $\Psi(\{z \in \mathbb{C} : |z| = 1\})$ and, conversely, can be reconstructed, along with the algorithm, from a given contour.

2. **Corner algorithms** (Section 4). A crucial role is played by contours that have a corner with external angle $\theta\pi, 1 < \theta < 2$. We prove that the respective algorithms accelerate the plain GD rate $O(t^{-\varsigma})$ to $O(t^{-\theta\varsigma})$. However, in Stochastic GD such algorithms have the negative effect of amplifying the sampling noise. By balancing these two effects, we establish the precise phase diagram of feasible accelerations of SGD under power-law spectral assumptions (Figure 1 right). In particular, we identify three natural sub-phases in the signal-dominated phase; in one of them acceleration up to $O(t^{-2\varsigma})$ is theoretically feasible.

3. **Implementation of Corner (S)GD** (Section 5). Ideal corner algorithms require an infinite memory, but can be fast approximated by finite-memory algorithms using fast rational approximations of the power function $z^\theta$. Experiments with a synthetic problem and MNIST confirm the practical acceleration.

## 2   Background

This section is largely based on the paper [29] to which we refer for details.

**Gradient descent with memory.**   Suppose that we wish to minimize a loss function $L(\mathbf{w})$ on a linear space $\mathcal{H}$. We consider gradient descent with size-$M$ memory that can be written as

$$\begin{pmatrix} \mathbf{w}_{t+1} - \mathbf{w}_t \\ \mathbf{u}_{t+1} \end{pmatrix} = \begin{pmatrix} -\alpha & \mathbf{b}^T \\ \mathbf{c} & D \end{pmatrix} \begin{pmatrix} \nabla L(\mathbf{w}_t) \\ \mathbf{u}_t \end{pmatrix}, \quad t = 0, 1, 2, \ldots \tag{1}$$

The vector $\mathbf{w}_t$ is the current step-$t$ approximation to an optimal vector $\mathbf{w}_*$, and $\mathbf{u}_t$ is an auxiliary vector representing the "memory" of the optimizer. These auxiliary vectors have the form $\mathbf{u} = (\mathbf{u}^{(1)}, \ldots \mathbf{u}^{(M)})^T$ with $\mathbf{u}^{(m)} \in \mathcal{H}$ and can be viewed as size-$M$ columns with each component belonging to $\mathcal{H}$. We refer to $M$ as the *memory size*. The parameter $\alpha$ (learning rate) is scalar, the parameters $\mathbf{b}, \mathbf{c}$ are $M$-dimensional column vectors, and $D$ is a $M \times M$ scalar matrix. The algorithm can be viewed as a sequence of transformations of size-$(M+1)$ column vectors $\begin{pmatrix} \mathbf{w}_t \\ \mathbf{u}_t \end{pmatrix}$ with $\mathcal{H}$-valued components. Throughout the paper, we only consider *stationary* algorithms, in the sense that the parameters $\alpha, \mathbf{b}, \mathbf{c}, D$ do not depend on $t$. The simplest nontrivial special case of GD with memory is Heavy Ball [19], in which $M = 1$ and $\mathbf{u}_t$ is the momentum.

Our theoretical results will rely on the assumption that $L$ is quadratic:

$$L(\mathbf{w}) = \frac{1}{2}\mathbf{w}^T \mathbf{H} \mathbf{w} - \mathbf{w}^T \mathbf{q}, \tag{2}$$

with a strictly positive definite $\mathbf{H}$. Throughout the paper, we will mostly be interested in infinite-dimensional Hilbert spaces $\mathcal{H}$, and we slightly abuse notation by interpreting $\mathbf{w}^T$ as the co-vector (linear functional $\langle \mathbf{w}, \cdot \rangle$) associated with vector $\mathbf{w}$. We will assume that $\mathbf{H}$ has a discrete spectrum with ordered strictly positive eigenvalues $\lambda_k \searrow 0$.

Let $\mathbf{w}_*$ be the optimal value of $L$ such that $\nabla L(\mathbf{w}_*) = \mathbf{H}\mathbf{w}_* - \mathbf{q} = 0$, and denote $\Delta\mathbf{w}_t = \mathbf{w}_t - \mathbf{w}_*$. Then, if $\Delta\mathbf{w}_t$ and $\mathbf{u}_t$ are eigenvectors of $\mathbf{H}$ with eigenvalue $\lambda$, then

$$\begin{pmatrix} \Delta\mathbf{w}_{t+1} \\ \mathbf{u}_{t+1} \end{pmatrix} = S_\lambda \begin{pmatrix} \Delta\mathbf{w}_t \\ \mathbf{u}_t \end{pmatrix}, \quad S_\lambda = \begin{pmatrix} 1 & \mathbf{b}^T \\ 0 & D \end{pmatrix} + \lambda \begin{pmatrix} -\alpha \\ \mathbf{c} \end{pmatrix} (1, \mathbf{0}^T), \tag{3}$$

and the new vectors $\Delta\mathbf{w}_{t+1}, \mathbf{u}_{t+1}$ are again eigenvectors of $\mathbf{H}$ with eigenvalue $\lambda$. As a result, performing the spectral decomposition of $\Delta\mathbf{w}_t, \mathbf{u}_t$ reduces the original dynamics (1) acting in $\mathcal{H} \otimes \mathbb{R}^{M+1}$ to a $\lambda$-indexed collection of independent dynamics each acting in $\mathbb{R}^{M+1}$.

For quadratic $L$, evolution (1) admits an equivalent representation

$$\mathbf{w}_{t+M+1} = \sum_{m=0}^{M} p_m \mathbf{w}_{t+m} + \sum_{m=0}^{M} q_m \nabla L(\mathbf{w}_{t+m}), \quad t = 0, 1, \ldots, \tag{4}$$

with constants $(p_m)_{m=0}^{M}, (q_m)_{m=0}^{M}$ such that $\sum_{m=0}^{M} p_m = 1$. These constants are found from the characteristic polynomial

$$\chi(\mu, \lambda) = \det(\mu - S_\lambda) = P(\mu) - \lambda Q(\mu), \ P(\mu) = \mu^{M+1} - \sum_{m=0}^{M} p_m \mu^m, \ Q(\mu) = \sum_{m=0}^{M} q_m \mu^m. \tag{5}$$

**Batch SGD with memory.** In batch Stochastic Gradient Descent (SGD), it is assumed that the loss has the form $L(\mathbf{w}) = \mathbb{E}_{\mathbf{x} \sim \rho} \ell(\mathbf{x}, \mathbf{w})$, where $\rho$ is some probability distribution of data points $\mathbf{x}$ and $\ell(\mathbf{x}, \mathbf{w})$ is the loss at the point $\mathbf{x}$. In the algorithm (1), we replace $\nabla L$ by $\nabla L_{B_t}$, where $B_t$ is a random batch of $|B|$ points sampled from distribution $\rho$, and $\nabla L_B$ is the empirical approximation to $L$, i.e. $L_B(\mathbf{w}) = \frac{1}{|B|} \sum_{\mathbf{x} \in B} \ell(\mathbf{x}, \mathbf{w})$. The samples $B_t$ at different steps $t$ are independent.

We assume $\ell$ to have the quadratic form $\ell(\mathbf{x}, \mathbf{w}) = \frac{1}{2}(\mathbf{x}^T \mathbf{w} - y(\mathbf{x}))^2$ for some scalar target function $y(\mathbf{x})$. Here, the inner product $\mathbf{x}^T \mathbf{w}$ can be viewed as a linear model acting on the feature vector $\mathbf{x}$. By projecting to the subspace of linear functions, we can assume w.l.o.g. that the target function $y(\mathbf{x})$ is itself linear in $\mathbf{x}$, i.e. $f(\mathbf{x}) = \mathbf{x}^T \mathbf{w}_*$ with some optimal parameter vector $\mathbf{w}_*$. (Later we will slightly weaken this assumption to also cover *unfeasible* solutions $\mathbf{w}_*$.) Then the full loss is quadratic as in Eq. (2): $L(\mathbf{w}) = \mathbb{E}_{\mathbf{x} \sim \rho} \frac{1}{2}(\mathbf{x}^T \Delta\mathbf{w})^2 = \frac{1}{2}\Delta\mathbf{w}^T \mathbf{H} \Delta\mathbf{w}$, where $\Delta\mathbf{w} = \mathbf{w} - \mathbf{w}_*$ and the Hessian $\mathbf{H} = \mathbb{E}_{\mathbf{x} \sim \rho}[\mathbf{x}\mathbf{x}^T]$.

**Mean loss evolution, SE approximation, and the propagator expansion.** Since the trajectory $\mathbf{w}_t$ in SGD is random, it is convenient to study the deterministic trajectory of batch-averaged losses $L_t = \mathbb{E}_{B_1, \ldots, B_{t-1}} L(\mathbf{w}_t)$. The sequence $L_t$ can be described exactly in terms of the second moments of $\mathbf{w}_t, \mathbf{u}_t$ that admit exact evolution equations. An important aspect of this evolution is that it involves 4'th order moments of the data distribution $\rho$ and so cannot in general be solved using only the second-order information available in the Hessian $\mathbf{H} = \mathbb{E}_{\mathbf{x} \sim \rho}[\mathbf{x}\mathbf{x}^T]$.

A convenient approach to handle this difficulty is the *Spectrally-Expressible (SE) approximation* proposed in [25]. It consists in assuming that there exist constants $\tau_1, \tau_2$ such that for all positive definite operators $\mathbf{C}$ in $\mathcal{H}$

$$\mathbb{E}_{\mathbf{x} \sim \rho}[\mathbf{x}\mathbf{x}^T \mathbf{C}\mathbf{x}\mathbf{x}^T] \approx \tau_1 \operatorname{Tr}[\mathbf{H}\mathbf{C}]\mathbf{H} - (\tau_2 - 1)\mathbf{H}\mathbf{C}\mathbf{H}. \tag{6}$$

In fact, this approximation holds *exactly* for some natural types of distribution $\rho$ (translation-invariant, gaussian). Otherwise, if the r.h.s. is only an upper or lower bound for the l.h.s., this implies a respective relation between the actual losses and the losses computed under the SE approximation. Theoretical predictions obtained under assumption (6) show good quantitative agreement with experiment on real data. We refer to [25, 29] for further discussion of the SE approximation.

The main benefit of the SE approximation is that it allows to write a convenient loss expansion

$$L_t = \frac{1}{2}\Big( V_{t+1} + \sum_{m=1}^{t} \sum_{0 < t_1 < \ldots < t_m < t+1} U_{t+1-t_m} U_{t_m - t_{m-1}} U_{t_{m-1} - t_{m-2}} \cdots U_{t_2 - t_1} V_{t_1} \Big) \tag{7}$$

with scalar *noise propagators* $U_t$ and *signal propagators* $V_t$. The signal propagators describe the error reduction during optimization in the absence of sampling noise, while the noise propagators describe the perturbing effect of sampling noise injected at times $t_1, \ldots, t_m$.

For our main results in Sections 3, 4, we will assume that $\tau_2 = 0$, implying particularly simple formulas for $U_t, V_t$:

$$U_t = \frac{\tau_1}{|B|} \sum_{k=1}^{\infty} \lambda_k^2 |(_1 \ \mathbf{o}^T) S_\lambda^{t-1} (_{\mathbf{c}}^{-\alpha})|^2, \quad V_t = \sum_{k=1}^{\infty} \lambda_k (\mathbf{e}_k^T \mathbf{w}_*)^2 |(_1 \ \mathbf{o}^T) S_\lambda^{t-1} (_{\mathbf{0}}^1)|^2, \quad (8)$$

where $\mathbf{e}_k$ is a normalized eigenvector for $\lambda_k$, and it is also assumed that optimization starts from $\mathbf{w}_0 = 0$ so that $\Delta \mathbf{w}_0 = \mathbf{w}_0 - \mathbf{w}_* = -\mathbf{w}_*$.

Importantly, the batch size $|B|$ affects $L_t$ only through the denominator in the coefficient in $U_t$. The deterministic GD corresponds to the limit $|B| \to \infty$: in this limit $U_t \equiv 0$ and $L_t = \frac{1}{2} V_{t+1}$.

**Convergence/divergence regimes.** Given expansion (7), we can deduce various convergence properties of the loss from the properties of the propagators $V_t, U_t$.

**Theorem 1** ([29])**.** *Let numbers $L_t$ be given by expansion (7) with some $U_t \geq 0, V_t \geq 0$. Let $U_\Sigma = \sum_{t=1}^{\infty} U_t$ and $V_\Sigma = \sum_{t=1}^{\infty} V_t$.*

1. *[Convergence] Suppose that $U_\Sigma < 1$. At $t \to \infty$, if $V_t = O(1)$ (respectively, $V_t = o(1)$), then also $L_t = O(1)$ (respectively, $L_t = o(1)$).*

2. *[Divergence] If $U_\Sigma > 1$ and $V_t > 0$ for at least one $t$, then $\sup_{t=1,2,\dots} L_t = \infty$.*

3. *[Signal-dominated regime] Suppose that there exist constants $\xi_V, C_V > 0$ such that $V_t = C_V t^{-\xi_V}(1 + o(1))$ as $t \to \infty$. Suppose also that $U_\Sigma < 1$ and $U_t = O(t^{-\xi_U})$ with some $\xi_U > \max(\xi_V, 1)$. Then*

$$L_t = \frac{C_V}{2(1 - U_\Sigma)} t^{-\xi_V}(1 + o(1)). \quad (9)$$

4. *[Noise-dominated regime] Suppose that there exist constants $\xi_V > \xi_U > 1, C_U > 0$ such that $U_t = C_U t^{-\xi_U}(1 + o(1))$ and $V_t = O(t^{-\xi_V})$ as $t \to \infty$. Let also that $U_\Sigma < 1$. Then*

$$L_t = \frac{V_\Sigma C_U}{2(1 - U_\Sigma)^2} t^{-\xi_U}(1 + o(1)). \quad (10)$$

**Spectral power laws.** The detailed convergence results in items 3, 4 of Theorem 1 require us to know the asymptotics of the propagators $U_t, V_t$. To this end we introduce power-law spectral assumptions on the eigenvalues and eigencomponents of $\mathbf{w}_*$ in our optimization problem:

$$\lambda_k = \Lambda k^{-\nu}(1 + o(1)), \quad k \to \infty, \quad (11)$$

$$\sum_{k:\lambda_k < \lambda} \lambda_k (\mathbf{e}_k^T \mathbf{w}_*)^2 = Q \lambda^\zeta (1 + o(1)), \quad \lambda \searrow 0, \quad (12)$$

with some constants $\Lambda, Q > 0$ and exponents $\nu > 0, \zeta > 0$. Such power laws are common in kernel methods or overparameterized models, and can be derived theoretically or observed empirically [1, 2, 3, 7, 10, 26, 27]. Conditions (11), (12) (or their weaker, inequality forms) are usually referred to as the *capacity* and *source* conditions, respectively [9]. The exponent $\zeta$ is akin to an inverse effective condition number: lower $\zeta$ means that the target and the solution have a heavier spectral tail of eigencomponents with small $\lambda$, making the problem harder. The exponent $\nu$ is akin to an inverse effective dimensionality of the problem: lower $\nu$ means a larger number of eigenvectors above a given spectral parameter $\lambda$. Only the source condition (12) matters for the non-stochastic GD rates, but in SGD the capacity condition (11) also becomes important due to the sampling noise.

If $0 < \zeta < 1$, then the source condition (12) is inconsistent with $\mathbf{w}_*$ having a finite $\mathcal{H}$-norm, i.e., strictly speaking, $\mathbf{w}_*$ is not an element of $\mathcal{H}$. Such a solution is called *unfeasible*. In fact, unfeasible scenarios are quite common both theoretically and in practice (see Section F). The Corner SGD to be proposed in Section 4 will be especially suitable for unfeasible scenarios. Note also that if $\nu < \frac{1}{2}$, then $U_1 = \infty$ and so $L_t \equiv \infty$, i.e. the loss immediately diverges.

**Stability and asymptotics of the propagators.** Let us say that a square matrix $A$ is *strictly stable* if all its eigenvalues are less than 1 in absolute value. It is natural to require the matrices $S_\lambda$ to be strictly stable for all $\lambda \in \text{spec}(\mathbf{H})$, since otherwise $U_t, V_t$, and hence $L_t$, will not generally even converge to 0 as $t \to \infty$. At $\lambda = 0$ the matrix $S_{\lambda=0}$ has eigenvalue 1 and additionally the eigenvalues of the matrix $D$; accordingly, we will assume that $D$ is strictly stable.

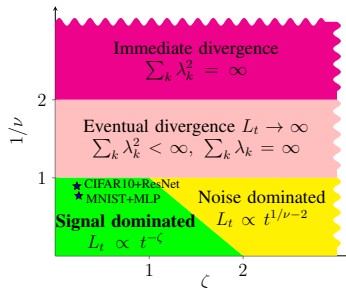 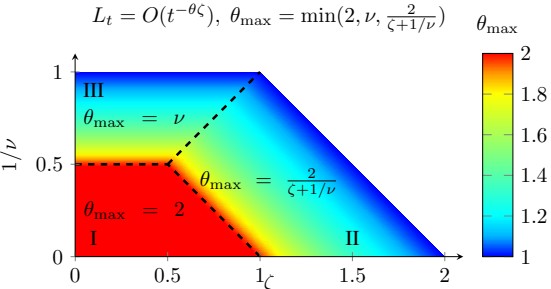

Figure 1: **Left:** The phase diagram of stationary finite-memory SGD from [25, 29]. **Right:** Maximum acceleration factor $\theta_{\max} = \min(2, \nu, \frac{2}{\zeta+1/\nu})$ for Corner SGD in the signal-dominated regime (see Theorem 4).

**Theorem 2** ([29]). *Suppose that $D$ and $S_\lambda$ are strictly stable for all $\lambda \in \mathrm{spec}(\mathbf{H})$. Recalling the characteristic polynomial $\chi(\mu, \lambda) = \det(\mu - S_\lambda) = P(\mu) - \lambda Q(\mu)$, define the effective learning rate*

$$\alpha_{\mathrm{eff}} = -Q(1) \Big/ \frac{dP}{d\mu}(1), \tag{13}$$

*and assume that $\alpha_{\mathrm{eff}} > 0$. Then, under spectral assumptions* (11), (12) *with $\nu > \frac{1}{2}$, the propagators $V_t, U_t$ given by Eq.* (8) *obey, as $t \to \infty$,*

$$V_t = (1 + o(1))Q\Gamma(\zeta + 1)(2\alpha_{\mathrm{eff}} t)^{-\zeta}, \tag{14}$$

$$U_t = (1 + o(1))\frac{(\alpha_{\mathrm{eff}}\Lambda)^{1/\nu}\tau_1\Gamma(2 - 1/\nu)}{|B|\nu}(2t)^{1/\nu-2}. \tag{15}$$

Combined with Theorem 1, this result yields the $(\zeta, 1/\nu)$-phase diagram shown in Figure 1 left. In particular, the region $\nu > 1, 0 < \zeta < 2 - 1/\nu$ represents the signal-dominated phase in which the noise effects are relatively weak and the loss convergence $L_t \propto t^{-\zeta}$ has the same exponent $\zeta$ as plain deterministic GD. This holds for all stationary finite-$M$ algorithms and so such algorithms cannot accelerate the exponent. In the present paper we will focus on the signal-dominated phase and propose an "infinite-memory" generalization of SGD that does accelerate the exponent.

## 3  The contour view of generalized (S)GD

We consider the propagator expansion (7) as a basis for our arguments. Observe that we can write the expression $\begin{pmatrix} 1 & \mathbf{0}^T \end{pmatrix} S_\lambda^t \begin{pmatrix} -\alpha \\ \mathbf{c} \end{pmatrix}$ appearing in the definition of propagator $U_t$ in Eq. (8) as

$$\begin{pmatrix} 1 & \mathbf{0}^T \end{pmatrix} S_\lambda^t \begin{pmatrix} -\alpha \\ \mathbf{c} \end{pmatrix} = \frac{1}{2\pi i} \oint_{|\mu|=r} \mu^t \begin{pmatrix} 1 & \mathbf{0}^T \end{pmatrix} (\mu - S_\lambda)^{-1} \begin{pmatrix} -\alpha \\ \mathbf{c} \end{pmatrix} d\mu, \tag{16}$$

where $|\mu| = r$ is a contour in the complex plane encircling all the eigenvalues of $S_\lambda$. Next, simple calculation (see Section A) shows that

$$\begin{pmatrix} 1 & \mathbf{0}^T \end{pmatrix} (\mu - S_\lambda)^{-1} \begin{pmatrix} -\alpha \\ \mathbf{c} \end{pmatrix} = \frac{Q(\mu)}{P(\mu) - \lambda Q(\mu)} = \frac{1}{\frac{P(\mu)}{Q(\mu)} - \lambda} = \frac{1}{\Psi(\mu) - \lambda}, \tag{17}$$

where $P(\mu) - \lambda Q(\mu)$ is the characteristic polynomial of $S_\lambda$ introduced in Eq. (5), and

$$\Psi(\mu) = \frac{P(\mu)}{Q(\mu)}. \tag{18}$$

We see, in particular, that the propagators $U_t$ depend on the algorithm parameters only through the function $\Psi$:

$$U_t = \frac{\tau_1}{|B|}\sum_{k=1}^{\infty}\lambda_k^2\left|\frac{1}{2\pi i}\oint_{|\mu|=r}\frac{\mu^{t-1}d\mu}{\Psi(\mu) - \lambda}\right|^2. \tag{19}$$

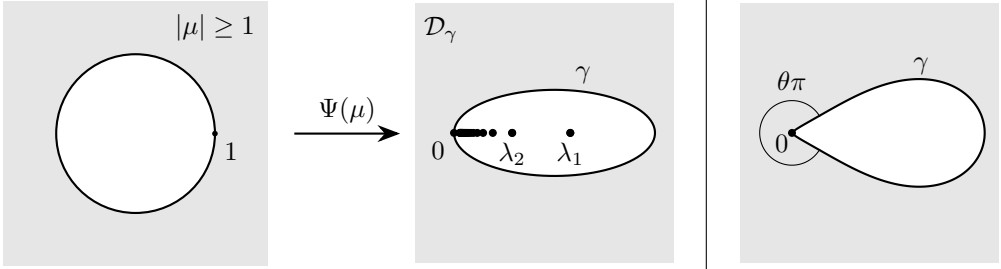

Figure 2: **Left:** The map $\Psi = \frac{P}{Q}$ for Heavy Ball with $P(\mu) = (\mu - 1)(\mu - 0.4)$ and $Q(\mu) = -\mu$. The contour $\gamma = \Psi(\{\mu : |\mu| = 1\})$ encircles $\mathrm{spec}(\mathbf{H})$. The map $\Psi$ bijectively maps $\{|\mu| > 1\}$ to the exterior open domain $\mathcal{D}_\gamma$ with boundary $\gamma$. See Sec. B for more examples and a general discussion of memory-1 contours. **Right:** Contour $\gamma$ corresponding to a corner map $\Psi$ with angle $\theta\pi$.

A similar observation can also be made regarding the propagators $V_t$. Indeed, $V_t$'s are different from $U_t$'s in that they involve the expression $(\,_1\, \mathbf{0}^T\,)S_\lambda^t(\,_{\mathbf{0}}^1\,)$ instead of $(\,_1\, \mathbf{0}^T\,)S_\lambda^t(\,_{\mathbf{c}}^{-\alpha}\,)$. The contour representation for $(\,_1\, \mathbf{0}^T\,)S_\lambda^t(\,_{\mathbf{0}}^1\,)$ is similar to Eq. (16), and then a simple calculation gives

$$(\,_1\, \mathbf{0}^T\,)(\mu - S_\lambda)^{-1}(\,_{\mathbf{0}}^1\,) = \frac{\Psi(\mu)}{(\Psi(\mu) - \lambda)(\mu - 1)}. \tag{20}$$

As a result,

$$V_t = \sum_{k=1}^\infty \lambda_k(\mathbf{e}_k^T \mathbf{w}_*)^2 \Big| \frac{1}{2\pi i} \oint_{|\mu|=r} \frac{\mu^{t-1}\Psi(\mu)d\mu}{(\Psi(\mu) - \lambda)(\mu - 1)} \Big|^2. \tag{21}$$

Recall from Eqs. (4),(5) that $P$ can be any monic polynomial (i.e., with leading coefficient 1) of degree $M + 1$ such that $P(1) = 0$, while $Q$ can be any polynomials of degree not greater than $M$. Since by Eq. (7) the loss trajectory $L_t$ is completely determined by the propagators $U_t, V_t$, we see that designing a stationary SGD with memory is essentially equivalent to designing a rational function $\Psi$ subject to these simple conditions. By (4), the function $\Psi = \frac{P}{Q}$ can be interpreted as describing the (frequency) response of the gradient sequence $(\nabla L(\mathbf{w}_t))$ to the sequence $(\mathbf{w}_t)$.

Let us consider the map $\Psi$ from the stability perspective. Recall that we expect $S_{\lambda_k}$ to be strictly stable for all the eigenvalues $\lambda_k \in \mathrm{spec}(\mathbf{H})$. In terms of $\Psi = \frac{P}{Q}$ this means that $\Psi(\mu) \neq \lambda_k$ for all $\mu \in \mathbb{C}$ such that $|\mu| \geq 1$. This shows, in particular, that we can set the radius $r = 1$ in Eqs. (19), (21). Additionally, if $D$ is strictly stable, then $S_0$ has only one simple eigenvalue of unit absolute value, $\mu = 1$, and so $\Psi(\mu) \neq 0$ for $|\mu| = 1, \mu \neq 1$. Let us introduce the curve $\gamma$ as the image of the unit circle under the map $\Psi$. Then the last condition means that the curve $\gamma$ goes through the point $0$ only once, at $\mu = 1$.

In general, the curve $\gamma$ can have a complicated shape with self-intersections, and the map $\Psi$ may not be injective on the domain $|\mu| \geq 1$. In particular, the singularity of $\Psi$ at $\mu = \infty$ is $\propto \mu^{M+1-\deg(Q)}$, so in a vicinity of $\mu = \infty$ the function $\Psi$ is injective if and only if $\deg(Q) = M$ (and in general $\Psi$ may also have other singularities at $|\mu| > 1$). However, we may expect natural, non-degenerate algorithms to correspond to simple non-intersecting curves $\gamma$ and injective maps $\Psi$ on $|\mu| \geq 1$. For example, this is the case for plain (S)GD and Heavy Ball, where $\gamma$ is a circle and an ellipse, respectively (Fig. 2 left). See Section B for a general discussion of memory-1 algorithms.

Given a non-intersecting (Jordan) contour $\gamma$, denote by $\mathcal{D}_\gamma$ the respective exterior open domain. Then, by Riemann mapping theorem, there exists a bijective holomorphic map $\Psi_\gamma : \{\mu \in \mathbb{C} : |\mu| > 1\} \to \mathcal{D}_\gamma$. Additionally, by Carathéodory's theorem[1] (see e.g. [11], p. 13) this map extends continuously to the boundary, $\Psi_\gamma : \{\mu \in \mathbb{C} : |\mu| = 1\} \to \gamma$. Such maps $\Psi_\gamma$ are non-unique, forming a three-parameter family $\Psi_\gamma \circ f$, where $f$ is a conformal automorphism of $\{\mu \in \mathbb{C} : |\mu| > 1\}$. However, recall that our maps $\Psi = \frac{P}{Q}$ had the properties $\Psi(\infty) = \infty$ and $\Psi(1) = 0$. These two requirements for $\Psi_\gamma$ uniquely fix the conformal isomorphism and hence $\Psi_\gamma$.

---

[1]Carathéodory's theorem considers bounded domains, but our domains $\{\mu \in \mathbb{C} : |\mu| > 1\}$ and $\mathcal{D}_\gamma$ are conformally isomorphic to bounded ones by simple transformations $z = 1/(\mu - \mu_0)$.

This suggests the following reformulation of the design problem for stationary SGD with memory. Rather than starting with the algorithm in the matrix or sequential forms (1), (4), we start with a contour $\gamma$ or the associated Riemann map $\Psi_\gamma$, and ensure a fast decay of the respective propagators $U_t, V_t$ given by (19), (21) (and hence, by Theorem 1, of the loss $L_t$). Of course, the resulting map $\Psi_\gamma$ will not be rational in general, but we can subsequently approximate it with a rational function $\frac{P}{Q}$ and in this way approximately reconstruct the algorithm.

## 4 Corner algorithms

To motivate the algorithms introduced in this section, observe from Eqs. (9), (14) that in the signal-dominated regime of stationary memory-$M$ SGD, we can decrease the coefficient $C_L$ in the asymptotic formula $L_t = (1 + o(1))C_L t^{-\zeta}$ by increasing $\alpha_{\mathrm{eff}}$ while keeping the total noise coefficient $U_\Sigma < 1$. Since $\Psi(1) = 0$, $\alpha_{\mathrm{eff}}$ can be reformulated in terms of $\Psi$ as

$$\alpha_{\mathrm{eff}} = -\frac{Q(1)}{\frac{dP}{d\mu}(1)} = -\Big(\frac{d\Psi}{d\mu}(1)\Big)^{-1}. \tag{22}$$

Thus, increasing $\alpha_{\mathrm{eff}}$ means making $-\frac{d\Psi}{d\mu}(1)$ a possibly smaller positive number. Regarding $U_\Sigma = \sum_{t=1}^{\infty} U_t$, note first that, by (19), it can be written as

$$U_\Sigma = \frac{\tau_1}{(2\pi)^2|B|} \sum_{k=1}^{\infty} \lambda_k^2 \sum_{t=1}^{\infty} \Big| \oint_{|\mu|=1} \frac{\mu^{t-1}d\mu}{\Psi(\mu) - \lambda} \Big|^2 = \frac{\tau_1}{(2\pi)^2|B|} \sum_{k=1}^{\infty} \lambda_k^2 \int_{-\pi}^{\pi} \frac{d\phi}{|\Psi(e^{i\phi}) - \lambda_k|^2}. \tag{23}$$

Indeed, since the function $(\Psi(\mu) - \lambda)^{-1}$ is holomorphic in $\{|\mu| > 1\}$ and vanishes as $\mu \to \infty$, the integrals $\oint$ here vanish for all nonpositive integers $t = 0, -1, -2, \ldots$ so that $\sum_t$ collapses to the squared $L^2$ norm by Parseval's identity. If the resulting series (23) converges, we can always ensure $U_\Sigma < 1$ by making the batch size $|B|$ large enough.

It is then natural to try $\Psi = \Psi_\gamma$ with a contour $\gamma$ having a corner at 0 with a particular angle. Denote the angle by $\theta\pi$ when measured in the external domain $\mathcal{D}_\gamma$ (Figure 2 right). Such contours correspond to maps $\Psi : \{|\mu| > 1\} \to \mathcal{D}_\gamma$ such that

$$\Psi(\mu) = -c_\Psi(\mu - 1)^\theta(1 + o(1)), \quad \mu \to 1, \tag{24}$$

with the standard branch of $(\mu - 1)^\theta$ and some constant $c_\Psi > 0$. We will refer to such $\Psi$ as *corner maps* and to the respective generalized SGD as *corner algorithms*. Formally,

$$-\frac{d\Psi}{d\mu}(\mu = 1) \sim c\theta(\mu - 1)^{\theta-1}|_{\mu=1+} = \begin{cases} +\infty, & \theta < 1 \\ +0, & \theta > 1 \end{cases} \tag{25}$$

so we are interested in $\theta > 1$. At the same time, we cannot take $\theta > 2$, since this would violate the stability condition $\Psi\{|\mu| > 1\} \cap \mathrm{spec}(\mathbf{H}) = \varnothing$. Thus, the relevant range of values for $\theta$ is $[1, 2]$. Within this range, increasing $\theta$ should have a positive $\alpha_{\mathrm{eff}}$-related effect but a negative $U_\Sigma$-related effect, since the contour $\gamma = \Psi(|\mu| = 1)$ is getting closer to the spectral segment $[0, \lambda_{\max}]$, thus amplifying the singularity $|\Psi(e^{i\phi}) - \lambda_k|^{-2}$ in Eq. (23). Our main technical result is

**Theorem 3** (C). *Let $\Psi$ be a holomorphic function in $\{\mu \in \mathbb{C} : |\mu| > 1\}$ commuting with complex conjugation and obeying power law condition (24) with some $1 < \theta < 2$. Assume that $\Psi$ extends continuously to a $C^1$ function on the closed domain $|\mu| \geq 1$, $\Psi(\mu) \to \infty$ as $\mu \to \infty$, and $\frac{d}{d\mu}\Psi(\mu) = O(|\mu - 1|^{\theta-1})$ as $\mu \to 1$. Assume also that $\Psi(\{\mu \in \mathbb{C} : |\mu| \geq 1, \mu \neq 1\}) \cap [0, \lambda_{\max}] = \varnothing$, where $\lambda_{\max} = \lambda_1$ is the largest eigenvalue of $\mathbf{H}$. Let power-law spectral assumptions (11),(12) hold with some $\nu > 1, 0 < \zeta < 2$. Then propagators (19), (21) obey the following $t \to \infty$ asymptotics.*

  *1. (Noise propagators) $U_t = C_U t^{\theta/\nu-2}(1 + o(1))$, with the coefficient*

  $$C_U = \frac{\tau_1}{|B|} \Lambda^{1/\nu} \int_\infty^0 r^2 F_U^2(r) dr^{-\theta/\nu} < \infty, \quad F_U(r) = \frac{1}{2\pi i} \int_{i\mathbb{R}} \frac{e^{rz}dz}{c_\Psi z^\theta + 1}.$$

  *2. (Signal propagators) $V_t = C_V t^{-\theta\zeta}(1 + o(1))$, with the coefficient*

  $$C_V = Q \int_0^\infty F_V^2(r) dr^{\theta\zeta} < \infty, \quad F_V(r) = \frac{1}{2\pi i} \int_{i\mathbb{R}} \frac{c_\Psi z^{\theta-1} e^{rz} dz}{c_\Psi z^\theta + 1}.$$

We see that the leading $t \to \infty$ asymptotics of the propagators are completely determined by the $\lambda \searrow 0$ spectral asymptotics of the problem and the $\mu \to 1$ singularity of the map $\Psi$. The functions $F_U, F_V$ can be written in terms of the Mittag-Leffler functions $E_{\theta,\theta}, E_\theta$ (see Section C).

Availability of the coefficients $C_U, C_V$ ensures that the leading asymptotics of $U_t, V_t$ are strict power laws with specific exponents $2 - \theta/\nu$ and $\theta\zeta$, respectively. Increasing $\theta$ indeed improves convergence of the signal propagators, but degrades convergence of the noise propagators.

The largest acceleration of the loss exponent $\zeta$ possibly achievable with corner algorithms is by a factor $\theta$ arbitrarily close to 2, but in general it will be lower since, by Theorem 1, the exponent of $L_t$ is the lower of the exponents of $U_t$ and $V_t$; accordingly, the optimal $\theta$ is obtained by balancing the two exponents, i.e. setting $\theta\zeta = 2 - \theta/\nu$. Also, we need the noise exponent $2 - \theta/\nu$ to be $> 1$, since otherwise the total noise coefficient $U_\Sigma = \infty$ and $L_t$ diverges for any batch size $|B| < \infty$.

Combining these considerations, we get the phase diagram of feasible accelerations (Figure 1 right).

**Theorem 4.** *Consider a problem with power-law spectral conditions* (11),(12) *in the signal-dominated phase, i.e.* $\nu > 1, 0 < \zeta < 2 - 1/\nu$. *Let* $\theta_{\max}$ *denote the supremum of those* $\theta$ *for which there exists a corner algorithm and batch size* $B$ *such that* $L_t = O(t^{-\theta\zeta})$. *Then*

$$\theta_{\max} = \min\left(2, \nu, \frac{2}{\zeta + 1/\nu}\right). \tag{26}$$

The phase diagram thus has three regions:

I. **Fully accelerated**: $\theta_{\max} = 2$, achieved for $\nu > 2, 0 < \zeta < 1 - 1/\nu$.

II. **Signal/noise balanced**: $\theta_{\max} = \frac{2}{\zeta+1/\nu} < 2$, $\max(1/\nu, 1 - 1/\nu) < \zeta < 2 - 1/\nu$. The condition $1/\nu < \zeta$ ensures that $U_\Sigma$ is finite and less than 1 for $|B|$ large enough.

III. **Limited by $U_\Sigma$-finiteness**: $\theta_{\max} = \nu < 2, 1 < \nu < 2, 0 < \zeta < 1/\nu$. The signal exponent $\theta_{\max}\zeta$ is less than the noise exponent $2 - \theta_{\max}/\nu$, but increasing $\theta$ makes $U_\Sigma$ diverge.

## 5 Finite-memory approximations of corner algorithms

Though corner maps $\Psi$ are irrational, they can be efficiently approximated by rational functions. It was originally famously discovered by [17] that the function $|x|$ can by approximated by order-$M$ rational functions with error $O(e^{-c\sqrt{M}})$. This result was later refined in various ways. In particular, [12] establish a rational approximation with a similar error bound for general power functions $z \mapsto z^\theta$ on complex domains. For $\theta \in (0, 1)$, this is done by writing

$$z^\theta = \frac{\sin(\theta\pi)}{\theta\pi} \int_0^\infty \frac{z \, dt}{t^{1/\theta} + z} = \frac{\sin(\theta\pi)}{\theta\pi} \int_{-\infty}^\infty \frac{z e^{\theta\pi i/2 + s} ds}{e^{\pi i/2 + s/\theta} + z} \tag{27}$$

and then approximating the last integral by the trapezoidal rule with uniform spacing $h = \pi\sqrt{2\theta/M}$.

In our setting, we start by explicitly defining a $\theta$-corner map. This can be done in many ways; we find it convenient to set

$$\Psi(\mu) = -A\left(\int_0^1 \frac{d\delta^{2-\theta}}{\mu - 1 + \delta}\right)^{-1} \frac{\mu - 1}{\mu} = A\left((\theta - 2)\int_0^\infty \frac{e^{-(2-\theta)s} ds}{\mu - 1 + e^{-s}}\right)^{-1} \frac{\mu - 1}{\mu} \tag{28}$$

with a scaling parameter $A > 0$.

**Proposition 1** (D). *For any* $1 < \theta < 2$, *Eq.* (28) *defines a holomorphic map* $\Psi : \mathbb{C} \setminus [0, 1] \to \mathbb{C}$ *such that*

$$\Psi(\mu) = \begin{cases} -A\mu(1 + o(1)), & \mu \to \infty, \\ -\frac{A(2-\theta)\pi}{\sin((2-\theta)\pi)}(\mu - 1)^\theta(1 + o(1)), & \mu \to 1, \end{cases} \tag{29}$$

*where* $z^\theta$ *denotes the standard branch in* $\mathbb{C} \setminus (-\infty, 0]$. *Also,* $\Psi(\{|\mu| \geq 1\}) \cap (0, 2A] = \varnothing$.

Following [12], we approximate the last integral in Eq. (28) as

$$\int_0^\infty \phi(s) ds \approx h \sum_{m=1}^M \phi((m - \tfrac{1}{2})h), \quad h = \frac{l}{\sqrt{M}}, \tag{30}$$

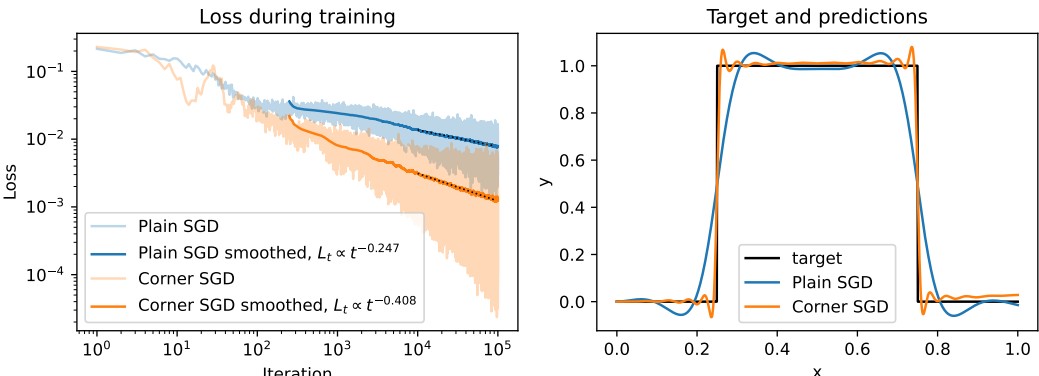

Figure 3: Training loss and final predictions of the kernel model (220) trained to fit the target $y(x) = \mathbf{1}_{[1/4,3/4]}(x)$ using either plain or corner SGD with batch size $|B| = 100$. The loss trajectories oscillate strongly, so their smoothed versions are also shown and used to estimate the exponents $\zeta$ in power laws $L_t \propto t^{-\zeta}$. Corner SGD has $\theta = 1.8$ and is approximated using finite memory $M = 5$ as in Proposition 2. We see that Corner SGD indeed accelerates the power-law convergence exponent of plain SGD. See Section F for details.

with some fixed constant $l$. Note that in contrast to (27), our integral and discretization are "one-sided" ($s > 0$), reflecting the fact that the corner map $\Psi(\mu)$ is power law only at $\mu \to 1$, which is related to the $s \to +\infty$ behavior of the integrand.

Let $\Psi^{(M)}$ denote the map $\Psi$ discretized with $M$ nodes by scheme (30). Observe that $\Psi^{(M)}$ is a rational function, $\Psi^{(M)} = \frac{P}{Q}$, where $\deg P = M + 1$ and $\deg Q \leq M$ (in particular, $P(\mu) = (\mu - 1)\prod_{m=1}^{M}(\mu - 1 + e^{-(m-1/2)h}))$. We can then associate to $\Psi^{(M)}$ a memory-$M$ algorithm (1) with particular $\alpha, \mathbf{b}, \mathbf{c}, D$, for example as follows.

**Proposition 2** (E). *Let $h = l/\sqrt{M}$ and*

$$D = \mathrm{diag}(1 - e^{-\frac{1}{2}h}, \dots, 1 - e^{-(M - \frac{1}{2})h}), \tag{31}$$

$$\mathbf{b} = (1, \dots, 1)^T, \tag{32}$$

$$\mathbf{c} = (c_1, \dots, c_M)^T, \quad c_m = A^{-1}(2 - \theta)he^{-(2-\theta)(m-1/2)h}(e^{-(m-1/2)h} - 1), \tag{33}$$

$$\alpha = A^{-1}(2 - \theta)h\frac{1 - e^{-(2-\theta)Mh}}{1 - e^{-(2-\theta)h}}e^{-(2-\theta)h/2}. \tag{34}$$

*Then the respective characteristic polynomial $\chi(\mu) = P(\mu) - \lambda Q(\mu)$ with $\frac{P}{Q} = \Psi^{(M)}$.*

Of course, as any stationary finite-memory algorithm, for very large $t$ the $M$-discretized corner algorithm can only provide a $O(t^{-\zeta})$ convergence of the loss. But, thanks to the $O(e^{-c\sqrt{M}})$ rational approximation bound, we expect that even with moderate $M$, for practically relevant finite ranges of $t$ the convergence should be close to $O(t^{-\theta\zeta})$ of the ideal corner algorithm.

Experiments with a synthetic problem and MNIST confirm that corner algorithms accelerate the exponents of plain SGD (see Appendix F and Figure 3). We also provide additional discussion of corner algorithms in Appendix G. In particular, we note that, while corner algorithms require significantly more memory than plain SGD, the amount of computation they perform is typically not much larger than for SGD. Our theoretical results significantly depended on the SE assumption (6) with $\tau_2 = 0$, but it appears that the theory can be extended to a more general setting (at the cost of more complicated expansions).

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

# Contents

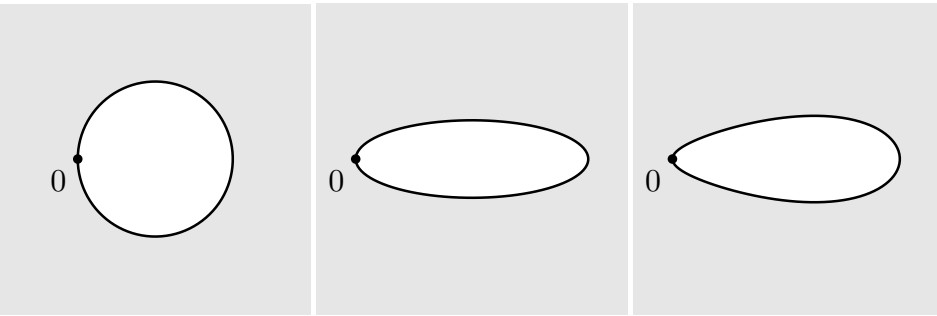

Figure 4: Contours $\gamma = \Psi(\{\mu : |\mu| = 1\})$ corresponding to different memory-1 maps $\Psi$ (see Section B). **Left:** plain Gradient Descent (a circle). **Center:** Heavy Ball (an ellipse; $\beta = 0.5$). **Right:** general memory-1 algorithms (a Zhukovsky airfoil; $\beta = 0.65, q_0 = 0.125, q_1 = -1$).

## A   Derivations of Section 3

We have

$$P(\mu) = \det(\mu - S_0) \tag{35}$$
$$= \det(\mu - S_\lambda + \lambda(\begin{smallmatrix} -\alpha \\ \mathbf{c} \end{smallmatrix})(\begin{smallmatrix} 1 & \mathbf{o}^T \end{smallmatrix})) \tag{36}$$
$$= \det(\mu - S_\lambda) \det\left(1 + \lambda(\begin{smallmatrix} -\alpha \\ \mathbf{c} \end{smallmatrix})(\begin{smallmatrix} 1 & \mathbf{o}^T \end{smallmatrix})(\mu - S_\lambda)^{-1}\right) \tag{37}$$
$$= (P(\mu) - \lambda Q(\mu))\left(1 + \lambda(\begin{smallmatrix} 1 & \mathbf{o}^T \end{smallmatrix})(\mu - S_\lambda)^{-1}(\begin{smallmatrix} -\alpha \\ \mathbf{c} \end{smallmatrix})\right). \tag{38}$$

It follows that

$$(\begin{smallmatrix} 1 & \mathbf{o}^T \end{smallmatrix})(\mu - S_\lambda)^{-1}(\begin{smallmatrix} -\alpha \\ \mathbf{c} \end{smallmatrix}) = \frac{1}{\lambda}\left(\frac{P(\mu)}{P(\mu) - \lambda Q(\mu)} - 1\right) \tag{39}$$
$$= \frac{Q(\mu)}{P(\mu) - \lambda Q(\mu)}. \tag{40}$$

Next, by Sherman-Morrison formula and the above identity,

$$(\mu - S_\lambda)^{-1} = (\mu - S_0 - \lambda(\begin{smallmatrix} -\alpha \\ \mathbf{c} \end{smallmatrix})(\begin{smallmatrix} 1 & \mathbf{o}^T \end{smallmatrix}))^{-1} \tag{41}$$
$$= (\mu - S_0)^{-1} + \lambda\frac{(\mu - S_0)^{-1}(\begin{smallmatrix} -\alpha \\ \mathbf{c} \end{smallmatrix})(\begin{smallmatrix} 1 & \mathbf{o}^T \end{smallmatrix})(\mu - S_0)^{-1}}{1 - \lambda(\begin{smallmatrix} 1 & \mathbf{o}^T \end{smallmatrix})(\mu - S_0)^{-1}(\begin{smallmatrix} -\alpha \\ \mathbf{c} \end{smallmatrix})} \tag{42}$$
$$= (\mu - S_0)^{-1} + \lambda\frac{(\mu - S_0)^{-1}(\begin{smallmatrix} -\alpha \\ \mathbf{c} \end{smallmatrix})(\begin{smallmatrix} 1 & \mathbf{o}^T \end{smallmatrix})(\mu - S_0)^{-1}}{1 - \lambda\frac{Q(\mu)}{P(\mu)}} \tag{43}$$

Using $(\begin{smallmatrix} 1 & \mathbf{o}^T \end{smallmatrix})(\mu - S_0)^{-1}(\begin{smallmatrix} 1 \\ \mathbf{0} \end{smallmatrix}) = \frac{1}{\mu-1}$, it follows that

$$(\begin{smallmatrix} 1 & \mathbf{o}^T \end{smallmatrix})(\mu - S_\lambda)^{-1}(\begin{smallmatrix} 1 \\ \mathbf{0} \end{smallmatrix}) = \frac{1}{\mu - 1} + \lambda\frac{\frac{Q(\mu)}{P(\mu)}\frac{1}{\mu-1}}{1 - \lambda\frac{Q(\mu)}{P(\mu)}} \tag{44}$$
$$= \frac{\frac{P(\mu)}{Q(\mu)}}{(\mu - 1)(\frac{P(\mu)}{Q(\mu)} - \lambda)}. \tag{45}$$

## B   Memory-1 contours

In figure 4 we show different contours $\gamma = \Psi(\{|\mu| = 1\})$ corresponding to memory-1 algorithms (see Section 3 for the introduction of contours). Below we discuss memory-1 algorithms and their contours in the order of increasing generality.

**Plain (S)GD.**   In (S)GD with learning rate $\alpha > 0$ we have $P(\mu) = \mu - 1$ and $Q(\mu) = -\alpha$, so

$$\Psi(\mu) = -\frac{\mu - 1}{\alpha}. \tag{46}$$

Thus, $\gamma$ is the circle $|z - \frac{1}{\alpha}| = \frac{1}{\alpha}$.

**Heavy Ball.** Heavy Ball with learning rate $\alpha$ and momentum parameter $\beta$ has standard stability conditions $\alpha > 0$, $\beta \in (-1, 1)$ and $\lambda_{\max} < \frac{2+2\beta}{\alpha}$ [20, 22]. We have $P(\mu) = (\mu - 1)(\mu - \beta)$ and $Q(\mu) = -\alpha\mu$, so

$$\Psi(\mu) = -\frac{(\mu - 1)(\mu - \beta)}{\alpha\mu}. \tag{47}$$

If $|\mu| = 1$, then $\mu\bar{\mu} = 1$ and hence

$$\Psi(\mu) = -\frac{1}{\alpha}(\mu + \beta\bar{\mu} - 1 - \beta). \tag{48}$$

Writing $\mu = x + iy$, we get

$$\Psi(\mu) = -\frac{1}{\alpha}((1 + \beta)x + i(1 - \beta)y - 1 - \beta). \tag{49}$$

It follows that $\gamma$ is an ellipse with the semi-axis $\frac{1+\beta}{\alpha}$ along $x$ and the semi-axis $\frac{1-\beta}{\alpha}$ along $y$. The learning rate $\alpha$ determines the size of the ellipse while the momentum parameter $\beta$ determines its shape. If $\beta > 0$, then the ellipse is elongated in the $x$ direction, and otherwise in the $y$ direction. Assuming $\beta > 0$, the eccentricity of the ellipse equals $e = \sqrt{1 - (1 - \beta)^2/(1 + \beta)^2} = \frac{2\sqrt{\beta}}{1+\beta}$. Plain GD is the special case of Heavy Ball with $\beta = 0$.

**General memory-1 (S)GD.** In a general memory-1 algorithm we have $P(\mu) = (\mu - 1)(\mu - \beta)$ and $Q(\mu) = q_0 + q_1\mu$, so

$$\Psi(\mu) = \frac{(\mu - 1)(\mu - \beta)}{q_0 + q_1\mu}. \tag{50}$$

Heavy Ball is the special case of general memory-1 algorithms with $q_0 = 0$.

In [29] it was shown that on the spectral interval $(0, \lambda_{\max}]$ the strict stability of the generalized memory-1 SGD is equivalent to the conditions

$$-1 < \beta < 1, \quad q_0 > -\frac{1 - \beta}{\lambda_{\max}}, \quad q_0 - \frac{2 + 2\beta}{\lambda_{\max}} < q_1 < -q_0 \tag{51}$$

(note that the Heavy Ball stability conditions result by setting $q_0 = 0, q_1 = -\alpha$).

**Zhukovsky airfoil representation.** The map $\Psi$ can be written as a composition of linear transformations and the Zhukovsky function

$$J(\mu) = \mu + \frac{1}{\mu}. \tag{52}$$

Indeed, let

$$\mu_1 \equiv f_1(\mu) \equiv q_0 + q_1\mu, \tag{53}$$

then

$$\Psi(\mu) = \frac{\left(\frac{\mu_1 - q_0}{q_1} - 1\right)\left(\frac{\mu_1 - q_0}{q_1} - \beta\right)}{\mu_1} \tag{54}$$

$$= \frac{\mu_1}{q_1^2} + \frac{r}{\mu_1} - \frac{2\frac{q_0}{q_1} + 1 + \beta}{q_1} \tag{55}$$

$$= \frac{\sqrt{r}}{q_1}J\left(\frac{\mu_1}{q_1\sqrt{r}}\right) - \frac{2\frac{q_0}{q_1} + 1 + \beta}{q_1}, \tag{56}$$

where

$$r = \left(\frac{q_0}{q_1} + 1\right)\left(\frac{q_0}{q_1} + \beta\right) \tag{57}$$

and $\sqrt{r}$ is imaginary if $r < 0$.

Thus, the contour $\gamma = \Psi(\{|\mu| = 1\})$ is a rescaled image of a circle under the Zhukovsky transform, i.e. a "Zhukovsky airfoil".

**Conditions of injectivity.** As discussed in Section 3, the case of maps $\Psi$ injective on the domain $|\mu| > 1$ seems especially natural and attractive. Let us examine when the map $\Psi$ given by Eq. (50) is injective. We can assume without loss that $q_1 \neq 0$ since otherwise the map $\Psi$ is not locally injective at $\infty$.

The Zhukovsky transform can be written as a composition of two linear fractional transformations and the function $w = z^2$:

$$J(\mu) = 2\frac{1+w}{1-w}, \quad w = z^2, \quad z = \frac{\mu-1}{\mu+1}. \tag{58}$$

The image of a generalized disc on the extended complex plane under a linear fractional map is again a generalized disc, and the map $w = z^2$ is injective on a generalized open disc if and only if the disc does not contain 0 and $\infty$. Hence, a necessary and sufficient condition for $J$ to be injective on a generalized open disc is that this disc not contain the points $\pm 1$. It follows that $\Psi$ is injective on the generalized disc $|\mu| > 1$ iff

$$\left| -\frac{q_0}{q_1} \pm \sqrt{r} \right| \leq 1. \tag{59}$$

Let us henceforth assume the stability condition $-1 < \beta < 1$ as given in Eq. (51). Consider separately the cases of negative and positive $r$.

1. $r \leq 0$ corresponds to $-1 \leq \frac{q_0}{q_1} \leq -\beta$. In this case condition (59) is equivalent to $-1 \leq \frac{q_0}{q_1}$, i.e. it holds.

   However, the special case $\frac{q_0}{q_1} = -1$ is the degenerate scenario in which the denominator of $\Psi$ vanishes at $\mu = 1$ and the stability condition $q_1 < -q_0$ in Eq. (51) is violated, so we will discard this special case.

2. $r > 0$ corresponds to $\frac{q_0}{q_1} < -1$ or $\frac{q_0}{q_1} > -\beta$. The option $\frac{q_0}{q_1} < -1$ is inconsistent with condition (59), leaving only the option $\frac{q_0}{q_1} > -\beta$.

   (a) If $\frac{q_0}{q_1} \leq 0$, then condition (59) is equivalent to

   $$\sqrt{r} \leq 1 + \frac{q_0}{q_1}, \tag{60}$$

   which holds true thanks to the assumption $\beta < 1$.

   (b) If $\frac{q_0}{q_1} \geq 0$, then condition (59) is equivalent to

   $$\sqrt{r} \leq 1 - \frac{q_0}{q_1}, \tag{61}$$

   which holds iff

   $$\frac{q_0}{q_1} \leq \frac{1-\beta}{3+\beta}. \tag{62}$$

Summarizing, assuming the stability condition $-1 < \beta < 1$ and excluding the degenerate case $q_0 = -q_1$, the condition of injectivity of the map $\Psi$ on the domain $|\mu| > 1$ reads

$$-1 < \frac{q_0}{q_1} \leq \frac{1-\beta}{3+\beta}. \tag{63}$$

We remark that this condition can also be reached in a different way. There are two obvious necessary conditions of injectivity of $\Psi$ on the set $|\mu| > 1$: the absence of poles of $\Psi$ and zeros of the derivative $\Psi'$ from this domain (the latter ensures the local injectivity). The absence of poles means that $-1 \leq \frac{q_0}{q_1} \leq 1$. The zeros of the derivative are given by the equation

$$\mu^2 + 2\frac{q_0}{q_1}\mu - (\beta+1)\frac{q_0}{q_1} - \beta = 0. \tag{64}$$

Both roots of a quadratic equation $\mu^2 + a\mu + b = 0$ lie inside the closed unit circle iff $|a| \leq 1+b \leq 2$. Applying this condition (and discarding the case $q_0/q_1 = -1$), we reach the same inequalities (63). In particular, the conditions of absence of poles and the roots of the derivative turn out to be not only necessary, but also sufficient.

**Algebraic equation of the contour.** The circle $|\mu| = 1$ is a real algebraic curve defined by the polynomial equation $x^2 + y^2 = 1$, where $\mu = x + iy$. Images of real algebraic curves under rational complex maps are again algebraic curves, and the corresponding equations can be found using, e.g., Macaulay resultants [21]. In the particular case of unit circle the computation can be performed in terms of standard resultants as follows.

Recall that $\Psi(\mu) = \frac{P(\mu)}{Q(\mu)}$, where $P$ is a polynomial of degree $M + 1$, and $Q$ is a polynomial of degree $\leq M$; we assume $P$ and $Q$ to have real coefficients. Denote $w = \Psi(\mu)$, then

$$wQ(\mu) = P(\mu). \tag{65}$$

Since $\mu$ belongs to the unit circle, $\mu\overline{\mu} = 1$. Applying complex conjugation and the identity $\overline{\mu} = 1/\mu$ to the above equation, we get the second equation

$$\overline{w}Q(1/\mu) = P(1/\mu). \tag{66}$$

Note that $\widetilde{Q}(\mu) = \mu^{M+1}Q(1/\mu)$ and $\widetilde{P}(\mu) = \mu^{M+1}P(1/\mu)$ are polynomials in $\mu$ of degree $M + 1$ or less. It follows that $\mu$ satisfies two polynomial conditions:

$$T_1(\mu) = 0, \quad T_2(\mu) = 0, \tag{67}$$

where

$$T_1(\mu) = P(\mu) - wQ(\mu), \quad T_2(\mu) = \widetilde{P}(\mu) - \overline{w}\widetilde{Q}(\mu), \tag{68}$$

i.e. $\mu$ is a common root of two polynomials, $T_1(\mu)$ and $T_2(\mu)$. Two polynomials have a common root iff their resultant vanishes. The polynomials $T_1(\mu), T_2(\mu)$ have degree $M + 1$ or less and include $w$ and $\overline{w}$ linearly in their coefficients. It follows that the set $\Psi(\{|\mu| = 1\})$ can be described by the equation

$$\mathrm{res}(T_1(\mu), T_2(\mu)) = 0, \tag{69}$$

which is a polynomial equation in $w$ and $\overline{w}$ of degree at most $2(M + 1)$.

We implement now this general program for $M = 1$. Given quadratic polynomials

$$T_1(\mu) = A\mu^2 + B\mu + C, \tag{70}$$
$$T_2(\mu) = D\mu^2 + E\mu + F, \tag{71}$$

their resultant can be written as

$$\mathrm{res}(T_1, T_2) = (AF - CD)^2 - (AE - BD)(BF - CE). \tag{72}$$

In our case

$$A = 1, \quad B = -(\beta + 1 + wq_1), \quad C = \beta - wq_0, \tag{73}$$
$$D = \beta - \overline{w}q_0, \quad E = -(\beta + 1 + \overline{w}q_1), \quad F = 1. \tag{74}$$

Considering real $\beta, q_0, q_1$ and $w = x + iy$, we get

$$\mathrm{res}(T_1, T_2) = (1 - (\beta - q_0x)^2 - q_0^2y^2)^2 \tag{75}$$
$$- (\beta^2 - 1 + [(q_1 - q_0)\beta - q_1 - q_0]x - q_0q_1(x^2 + y^2))^2 \tag{76}$$
$$- (\beta + 1)^2(q_0 + q_1)^2y^2. \tag{77}$$

It follows that the contour $\Psi(\{|\mu| = 1\})$ can be described by the quartic (in general) equation

$$(1 - (\beta - q_0x)^2 - q_0^2y^2)^2 = (\beta^2 - 1 + [(q_1 - q_0)\beta - q_1 - q_0]x - q_0q_1(x^2 + y^2))^2 \tag{78}$$
$$+ (\beta + 1)^2(q_0 + q_1)^2y^2. \tag{79}$$

As expected, in the Heavy Ball case $q_0 = 0$ this equation degenerates into the quadratic equation

$$(1 - \beta^2)^2 = (\beta^2 - 1 + (\beta - 1)q_1x)^2 + (\beta + 1)^2q_1^2y^2. \tag{80}$$

 # C  Proof of Theorem 3

 ## C.1  The noise propagators

 **The function $F_U$.**  Let us introduce the values

$$U(t,\lambda) = \frac{1}{2\pi i} \oint_{|\mu|=1} \frac{\mu^{t-1}d\mu}{\Psi(\mu)-\lambda} = \frac{1}{2\pi} \int_{-\pi}^{\pi} \frac{e^{it\phi}d\phi}{\Psi(e^{i\phi})-\lambda} \tag{81}$$

 so that, by Eq. (19), the propagator $U_t$ can be written as

$$U_t = \frac{\tau_1}{|B|} \sum_{k=1}^{\infty} \lambda_k^2 |U(t,\lambda)|^2. \tag{82}$$

 With the change of variables $\phi = s\lambda^{1/\theta}$,

$$U(t,\lambda) = \frac{-\lambda^{1/\theta-1}}{2\pi} \int_{-\pi/\lambda^{1/\theta}}^{\pi/\lambda^{1/\theta}} \frac{e^{it\lambda^{1/\theta}s}ds}{-\Psi(e^{is\lambda^{1/\theta}})/\lambda+1} = -\lambda^{1/\theta-1}F_U(t\lambda^{1/\theta},\lambda), \tag{83}$$

 where we have denoted

$$F_U(r,\lambda) = \frac{1}{2\pi} \int_{-\pi/\lambda^{1/\theta}}^{\pi/\lambda^{1/\theta}} \frac{e^{irs}ds}{-\Psi(e^{is\lambda^{1/\theta}})/\lambda+1}. \tag{84}$$

  Recall that we assume $\Psi(\mu) = -c_\Psi(\mu-1)^\theta(1+o(1))$ as $\mu \to 1$. By formally taking the limit $\lambda \searrow 0$ in the integral, we then expect $F_U(r,\lambda)$ to converge to

$$F_U(r,0) \stackrel{\text{def}}{=} F_U(r) \stackrel{\text{def}}{=} \frac{1}{2\pi} \int_{-\infty}^{\infty} \frac{e^{irs}ds}{c_\Psi e^{i(\text{sign } s)\theta\pi/2}|s|^\theta + 1} \tag{85}$$

 for any fixed $r$. This integral can be equivalently written as

$$F_U(r) = \frac{1}{2\pi i} \int_{i\mathbb{R}} \frac{e^{rz}dz}{c_\Psi z^\theta + 1}, \tag{86}$$

 assuming the standard branch of $z^\theta$ holomorphic in $\mathbb{C} \setminus (-\infty, 0]$.

     The function $F_U$ can be viewed (up to a coefficient) as the inverse Fourier transform of the function $s \mapsto (c_\Psi e^{i(\text{sign } s)\theta\pi/2}|s|^\theta + 1)^{-1}$. Note that, thanks to the condition $\theta > 1$, the latter function is Lebesgue-integrable, so $F_U(r)$ is well-defined and continuous for all $r \in \mathbb{R}$. The function $F_U$ can also be written in terms of the special Mittag-Leffler function $E_{\theta,\theta}$ (see its integral representation (6.8) in [13]):

$$F_U(r) = \frac{r^{\theta-1}}{c_\Psi} E_{\theta,\theta}\left(-\frac{r^\theta}{c_\Psi}\right), \quad E_{a,b}(z) = \frac{1}{2\pi i} \int_\gamma \frac{t^{a-b}e^t dt}{t^a - z}, \tag{87}$$

 where the integration path $\gamma$ encircles the cut $(-\infty, 0]$ and the singularities of the denominator.

  The following asymptotic properties of $F_U(r)$ can be derived from the general asymptotic expansions of Mittag-Leffler functions (sections 1 and 6 in [13]), but we provide proofs for completeness.

 **Lemma 1.**

     *1.* $F_U(r) = 0$ *for* $r \leq 0$.

     *2.* $F_U(r) = (1+o(1))\frac{1}{c_\Psi\Gamma(\theta)}r^{\theta-1}$ *as* $r \searrow 0$.

     *3.* $F_U(r) = (1+o(1))\frac{-c_\Psi}{\Gamma(-\theta)}r^{-\theta-1}$ *as* $r \to +\infty$.

    *Proof.* 1. Consider the function $f(z)$ integrated in Eq. (86). For any $r \in \mathbb{R}$ and $\theta \in (1,2)$, the function $f$ is holomorphic in any strip $\mathcal{T}_a = \{0 < \Re z < a\}, a > 0$, and is bounded in $\mathcal{T}_a$ as $|f(z)| = O(|z|^{-\theta})$. It follows that the integration line $i\mathbb{R}$ can be deformed to $i\mathbb{R} + a$ without changing the integral. If $r < 0$, then by letting $a \to +\infty$ we can make the integral arbitrarily small.

2. By the change of variables $rz = z'$,

$$F_U(r) = u(r)r^{\theta - 1}, \tag{88}$$

where

$$u(r) = \frac{1}{2\pi i c_\Psi} \int_{i\mathbb{R}} \frac{e^{z'} dz'}{z'^\theta + c_\Psi^{-1} r^\theta}. \tag{89}$$

We can find $\lim_{r \searrow 0} u(r)$ as follows. Observe that the integration line $i\mathbb{R}$ can be deformed to the line $\gamma_a, a > 0$, encircling the negative semi-axis:

$$\gamma_a = \gamma_{a,1} \cup \gamma_{a,2} \cup \gamma_{a,3}, \tag{90}$$
$$\gamma_{a,1} = \{z \in \mathbb{C} : \Im z = -a, \Re z \leq 0\}, \tag{91}$$
$$\gamma_{a,2} = \{z \in \mathbb{C} : |z| = a, -\tfrac{\pi}{2} < \arg z < \tfrac{\pi}{2})\}, \tag{92}$$
$$\gamma_{a,3} = \{z \in \mathbb{C} : \Im z = a, \Re z \leq 0\}. \tag{93}$$

Indeed, if $r$ is sufficiently small, then this deformation occurs within the holomorphy domain of the integrated function. The integral is preserved since $\theta > 0$ and since we deform in the half-plane where the argument of $e^{z'}$ has $\Re z' < 0$.

Thus, for any fixed $a > 0$ we have

$$\lim_{r \searrow 0} u(r) = \lim_{r \searrow 0} \frac{1}{2\pi i c_\Psi} \int_{\gamma_a} \frac{e^{z'} dz'}{z'^\theta + c_\Psi^{-1} r^\theta} = \frac{1}{2\pi i c_\Psi} \int_{\gamma_a} \frac{e^{z'} dz'}{z'^\theta} = \frac{1}{2\pi i c_\Psi (\theta - 1)} \int_{\gamma_a} \frac{e^{z'} dz'}{z'^{\theta - 1}}, \tag{94}$$

where in the last step we integrated by parts. In the last integral, thanks to the weakness of the singularity $z'^{1-\theta}$ at $z' = 0$ (note that $1 - \theta > -1$), we can let $a \to 0$:

$$\int_{\gamma_a} \frac{e^{z'} dz'}{z'^{\theta - 1}} = \int_0^{+\infty} e^{-s} s^{1-\theta} (e^{-\pi i(1-\theta)} - e^{\pi i(1-\theta)}) ds \tag{95}$$

$$= 2i \sin(\pi(\theta - 1)) \Gamma(2 - \theta) \tag{96}$$

$$= \frac{2\pi i}{\Gamma(\theta - 1)}, \tag{97}$$

where in the last step we used the identity $\Gamma(z)\Gamma(1 - z) = \frac{\pi}{\sin(\pi z)}$. This is essentially Hankel's representation of the Gamma function, valid for all $\theta \in \mathbb{C}$ by analytic continuation. Summarizing,

$$\lim_{r \searrow 0} u(r) = \frac{1}{c_\Psi (\theta - 1) \Gamma(\theta - 1)} = \frac{1}{c_\Psi \Gamma(\theta)}. \tag{98}$$

3. We start by performing integration by parts in $F_U$ :

$$F_U(r) = \frac{-1}{2\pi i r} \int_{i\mathbb{R}} e^{rz} d \frac{1}{c_\Psi z^\theta + 1} = \frac{c_\Psi \theta}{2\pi i r} \int_{i\mathbb{R}} \frac{e^{rz} z^{\theta - 1} dz}{(c_\Psi z^\theta + 1)^2}. \tag{99}$$

Performing again the change of variables $rz = z'$, we have

$$F_U(r) = v(r)r^{-\theta - 1}, \tag{100}$$

where

$$v(r) = \frac{c_\Psi \theta}{2\pi i} \int_{i\mathbb{R}} \frac{e^{z'} z'^{\theta - 1} dz'}{(c_\Psi (z'/r)^\theta + 1)^2}. \tag{101}$$

To compute $\lim_{r \to \infty} v(r)$, we again transform the integration line. Let $\gamma'$ be a line that lies in the domain $\mathbb{C} \setminus (-\infty, 0)$ and can be represented as the graph of a function $\Re z = f(\Im z)$ such that

$$f(y) \geq c_1 |y| - c_0 \tag{102}$$

with some constant $c_1 > 0$ and $c_0$.

Note that the integrated function has two singular points $z' \in \mathbb{C} \setminus (-\infty, 0]$ where the denominator $c_\Psi (z'/r)^\theta + 1 = 0$. These two points depend linearly on $r$. Require additionally that $\gamma'$ lie to the right of these points for all $r > 0$, so that $i\mathbb{R}$ can be deformed to $\gamma'$ without meeting the singularities.

522 This requirement is feasible with a small enough $c_1 > 0$ since, by the condition $\theta < 2$, the imaginary
523 parts of the singular points are negative.

524 With these assumptions, integration in Eq. (101) can be changed to integration over $\gamma'$. Thanks to
525 condition (102), the integrand converges exponentially fast at $z' \to \infty$, and we can take the limit
526 $r \to +\infty$ :

$$\lim_{r \to +\infty} v(r) = \frac{c_\Psi \theta}{2\pi i} \int_{\gamma'} e^{z'} z'^{\theta-1} dz'. \tag{103}$$

527 The contour $\gamma'$ can now be transformed to a contour encircling the negative semi-axis, and applying
528 Eq. (97) we get

$$\lim_{r \to +\infty} v(r) = \frac{c_\Psi \theta}{\Gamma(1-\theta)} = \frac{-c_\Psi}{\Gamma(-\theta)}. \tag{104}$$

529 $\qquad\qquad\qquad\qquad\qquad\qquad\qquad\qquad\qquad\qquad\qquad\qquad\qquad\qquad\qquad\qquad\qquad\qquad\qquad\square$

530 **The formal leading term in $U_t$.** We have

$$U_t = \frac{\tau_1}{|B|} \sum_{k=1}^\infty \lambda_k^2 |U(t, \lambda_k)|^2 = \frac{\tau_1}{|B|} \sum_k \lambda_k^{2/\theta} F_U^2(t\lambda_k^{1/\theta}, \lambda_k). \tag{105}$$

531 To extract the leading term in this expression, we set the second argument in $F_U(t\lambda_k^{1/\theta}, \lambda_k)$ to 0:

$$U_t^{(1)} \overset{\text{def}}{=} \frac{\tau_1}{|B|} \sum_k \lambda_k^{2/\theta} F_U^2(t\lambda_k^{1/\theta}) = \frac{\tau_1}{|B|} a_t t^{\theta/\nu-2}, \tag{106}$$

532 where

$$a_t = t^{2-\theta/\nu} \sum_k \lambda_k^{2/\theta} F_U^2(t\lambda_k^{1/\theta}) = t^{-\theta/\nu} \sum_k (t\lambda_k^{1/\theta})^2 F_U^2(t\lambda_k^{1/\theta}). \tag{107}$$

533

**Lemma 2.**

$$\lim_{t \to \infty} a_t = \Lambda^{1/\nu} \int_\infty^0 r^2 F_U^2(r) dr^{-\theta/\nu} = \Lambda^{1/\nu} \frac{\theta}{\nu} \int_0^\infty r^{1-\theta/\nu} F_U^2(r) dr < \infty. \tag{108}$$

534 *Proof.* Note first that the integral on the right is convergent. Indeed, by statement 2 of Lemma 1,
535 $r^{1-\theta/\nu} F^2(r) \propto r^{1-\theta/\nu+2(\theta-1)} = r^{\theta(2-1/\nu)-1}$ near $r = 0$. Since we assume $\nu > 1$ and $\theta > 1$, the
536 function $r^{1-\theta/\nu} F^2(r)$ is bounded near $r = 0$. Also, by statement 3 of Lemma 1, $r^{1-\theta/\nu} F^2(r) \propto$
537 $r^{1-\theta/\nu-2(\theta+1)} = O(r^{-3})$ as $r \to +\infty$.

538 For any interval $I$ in $\mathbb{R}_+$, denote by $S_{I,t}$ the part of the expansion (107) of $a_t$ corresponding to the
539 terms with $t\lambda_k^{1/\theta} \in I$ :

$$S_{I,t} = t^{-\theta/\nu} \sum_{k:t\lambda_k^{1/\theta} \in I} (t\lambda_k^{1/\theta})^2 F_U^2(t\lambda_k^{1/\theta}). \tag{109}$$

540 Recall that the eigenvalues $\lambda$ are ordered and $\lambda_k = \Lambda k^{-\nu}(1 + o(1))$ by capacity condition (11). It
541 follows that for a given fixed number $r > 0$, the condition $t\lambda_k^{1/\theta} > r$ holds whenever $k < k_r$, where

$$k_r = (1 + o(1))\Lambda^{1/\nu}(t/r)^{\theta/\nu}, \quad t \to \infty. \tag{110}$$

542 Then, for $I = [u, v]$ with $0 < u < v < \infty$ we have

$$\liminf_{t \to \infty} S_{I,t} \geq \Lambda^{1/\nu} \inf_{r \in I}[r^2 F_U^2(r)](u^{-\theta/\nu} - v^{-\theta/\nu}), \tag{111}$$

$$\limsup_{t \to \infty} S_{I,t} \leq \Lambda^{1/\nu} \sup_{r \in I}[r^2 F_U^2(r)](u^{-\theta/\nu} - v^{-\theta/\nu}). \tag{112}$$

543 Moreover, for any interval $I = [u, v]$ with $0 < u < v < \infty$ we can approximate $\int_I r^2 F_U^2(r) dr^{-\theta/\nu}$
544 by integral sums corresponding to sub-divisions $I = I_1 \cup I_2 \cup \ldots \cup I_n$, apply the above inequalities
545 to each $I_s$, and conclude that

$$\lim_{t \to \infty} S_{I,t} = \Lambda^{1/\nu} \int_I r^2 F_U^2(r) dr^{-\theta/\nu}. \tag{113}$$

It remains to handle the two parts of $a_t$ corresponding to the remaining intervals $I = [0, u]$ and $I = [v, \infty)$. It suffices to show that the associated contributions $S_{I,t}$ can be made arbitrarily small uniformly in $t$ by making $u$ small and $v$ large enough.

Consider first the interval $I = [v, \infty)$. Note that by Lemma 1 for all $r > 1$ we can write

$$r^2 F_U^2(r) \leq C r^{-2\theta} \tag{114}$$

with some constant $C$, and we also have for all $k$

$$\Lambda_- k^{-\nu} \leq \lambda_k \leq \Lambda_+ k^{-\nu} \tag{115}$$

for suitable constants $\Lambda_-, \Lambda_+$. It follows that

$$S_{I,t} \leq t^{-\theta/\nu} \sum_{k:t(\Lambda_+ k^{-\nu})^{1/\theta} > v} C(t(\Lambda_- k^{-\nu})^{1/\theta})^{-2\theta} \tag{116}$$

$$= t^{-\theta/\nu - 2\theta} C \Lambda_-^{-2} \sum_{k=1}^{\Lambda_+^{1/\nu}(t/v)^{\theta/\nu}} k^{2\nu} \tag{117}$$

$$= O(1) t^{-\theta/\nu - 2\theta} (t/v)^{(\theta/\nu)(2\nu+1)} \tag{118}$$

$$= O(1) v^{-(\theta/\nu)(2\nu+1)}, \tag{119}$$

with $O(1)$ denoting an expression bounded by a $t, v$-independent constant. This is the desired convergence property of $S_{I,t}$.

Similarly, for the other interval $I = [0, u]$ we use the inequality

$$r^2 F_U^2(r) \leq C r^{2\theta}, \quad r < 1, \tag{120}$$

also following by Lemma 1. Then

$$S_{I,t} \leq t^{-\theta/\nu} \sum_{k:t(\Lambda_- k^{-\nu})^{1/\theta} < u} C(t(\Lambda_+ k^{-\nu})^{1/\theta})^{2\theta} \tag{121}$$

$$= t^{-\theta/\nu + 2\theta} C \Lambda_+^2 \sum_{k=\Lambda_-^{1/\nu}(t/u)^{\theta/\nu}}^{\infty} k^{-2\nu} \tag{122}$$

$$= O(1) t^{-\theta/\nu + 2\theta} (t/u)^{(\theta/\nu)(1-2\nu)} \tag{123}$$

$$= O(1) u^{(\theta/\nu)(2\nu-1)}, \tag{124}$$

which is the desired convergence property of $S_{I,t}$ since $\nu > 1$. □

**Completion of proof.** We have shown that if we replace $F_U(t\lambda_k^{1/\theta}, \lambda_k)$ by $F_U(t\lambda_k^{1/\theta})$ in Eq. (105), we get desired asymptotics of $U_t$ in the limit $t \to +\infty$. We will show now that this replacement introduces a lower-order correction $o(t^{\theta/\nu-2})$; this will complete the proof.

We start with a technical lemma (to be applied with $f = \Psi$) giving a lower bound for deviations of asymptotic power law functions with $\theta < 2$ from real values.

**Lemma 3.** *Suppose that $f : \{\mu \in \mathbb{C} : |\mu| = 1\} \to \mathbb{C}$ is continuous, $f(\mu) = -c(\mu-1)^\theta(1+o(1))$ as $\mu \to 1$ with some $\theta \in [0, 2)$ and $c > 0$. Suppose also that $f(\{\mu \in \mathbb{C} : |\mu| = 1, \mu \neq 1\}) \cap [0, \lambda_{\max}] = \varnothing$ for some $\lambda_{\max} > 0$. Then there exist a constant $C > 0$ such that*

$$|f(e^{is}) - \lambda| \geq C(|s|^\theta + \lambda), \quad s \in [-\pi, \pi], \lambda \in [0, \lambda_{\max}]. \tag{125}$$

*Proof.* If we fix any small $\epsilon > 0$, then, by the condition $f(\{\mu \in \mathbb{C} : |\mu| = 1, \mu \neq 1\}) \cap [0, \lambda_{\max}] = \varnothing$ and a compactness argument, there exist $C', C > 0$ such that

$$|f(e^{is}) - \lambda| > C' > C(|s|^\theta + \lambda), \quad s \in [-\pi, -\epsilon] \cap [\epsilon, \pi], \lambda \in [0, \lambda_{\max}]. \tag{126}$$

567 It remains to establish inequality (125) for $|s| < \epsilon$. Since $f(\mu) = c(\mu-1)^\theta(1+o(1))$ and $\theta \in [0, 2)$,

$$|f(e^{is}) - \lambda| = |e^{i\operatorname{sign}(s)\theta\pi/2}c|s|^\theta(1+o(1)) + \lambda| \tag{127}$$

$$= |e^{i\operatorname{sign}(s)\theta\pi/4}c|s|^\theta(1+o(1)) + \lambda e^{-i\operatorname{sign}(s)\theta\pi/4}| \tag{128}$$

$$\geq \Re[e^{i\operatorname{sign}(s)\theta\pi/4}c|s|^\theta(1+o(1)) + \lambda e^{-i\operatorname{sign}(s)\theta\pi/4}] \tag{129}$$

$$= \cos(\theta\pi/4)(c|s|^\theta(1+o(1)) + \lambda) \tag{130}$$

$$\geq \tfrac{1}{2}\min(c, 1)\cos(\theta\pi/4)(|s|^\theta + \lambda) \tag{131}$$

568 for $|s|$ small enough. $\qquad\square$

569 **Lemma 4.**

570    *1. $|F_U(r, \lambda) - F_U(r)| = o(1)$ as $\lambda \to 0$, uniformly in all $r \in \mathbb{R}$.*

571    *2. $F_U(r, \lambda) = O(\frac{1}{r})$ for all $r$ of the form $r = t\lambda^{1/\theta}, t = 1, 2, \ldots$, uniformly in all $\lambda \in$*
572    $(0, \lambda_{\max}]$.

573 *Proof.* 1. It suffices to show that, as $\lambda \searrow 0$, the functions

$$f_\lambda(s) = -(2\pi)^{-1}(-\Psi(e^{is\lambda^{1/\theta}})/\lambda + 1)^{-1}\mathbf{1}_{[-\pi/\lambda^{1/\theta}, \pi/\lambda^{1/\theta}]}(s) \tag{132}$$

574 converge in $L^1(\mathbb{R})$ to

$$f_0(s) = -(2\pi)^{-1}(c_\Psi e^{i(\operatorname{sign} s)\theta\pi/2}|s|^\theta + 1)^{-1}. \tag{133}$$

575 Let us divide the interval $[-\pi/\lambda^{1/\theta}, \pi/\lambda^{1/\theta}]$ into two subsets:

$$I_1(\lambda) = [-\lambda^{-h}, \lambda^{-h}], \tag{134}$$

$$I_2(\lambda) = [-\pi/\lambda^{1/\theta}, \pi/\lambda^{1/\theta}] \setminus I_1(\lambda), \tag{135}$$

576 where $h$ is some fixed number such that $\frac{1}{\theta^2} < h < \frac{1}{\theta}$.

577 By Lemma 3, $|\Psi(e^{is\lambda^{1/\theta}})/\lambda - 1| \geq c|s|^\theta$ uniformly for all $s \in [-\pi/\lambda^{1/\theta}, \pi/\lambda^{1/\theta}]$ and $\lambda \in$
578 $(0, \lambda_{\max}]$. It follows that

$$\inf_{s \in I_2(\lambda)} |\Psi(e^{is\lambda^{1/\theta}})/\lambda - 1| \geq c\lambda^{-h\theta}, \quad \lambda \in (0, \lambda_{\max}], \tag{136}$$

579 for some constant $c > 0$. Using the condition $\frac{1}{\theta^2} < h$, it follows that

$$\int_{I_2(\lambda)} |f_\lambda(s)|ds = O(\lambda^{-1/\theta}\lambda^{h\theta}) = o(1), \quad \lambda \searrow 0. \tag{137}$$

580 Thus, we can assume without loss that the functions $f_\lambda$ vanish outside the intervals $I_1(\lambda)$. On these
581 intervals, thanks to the condition $h < \frac{1}{\theta}$, we have

$$f_\lambda(s) = -(2\pi)^{-1}(c_\Psi e^{i(\operatorname{sign} s)\theta\pi/2}|s|^\theta(1+o(1)) + 1)^{-1} \tag{138}$$

582 uniformly in $s \in I_1(\lambda)$. We can then apply the dominated convergence theorem to the functions
583 $|f_\lambda - f_0|$, with a dominating function $C(1 + |s|^\theta)^{-1}$, and conclude that $f_\lambda \to f_0$ in $L^1(\mathbb{R})$, as
584 desired.

585 2. We start by performing integration by parts in $U(t, \lambda)$:

$$U(t, \lambda) = \frac{1}{2\pi i t} \oint_{|\mu|=1} \frac{d\mu^t}{\Psi(\mu) - \lambda} = \frac{1}{2\pi i t} \oint_{|\mu|=1} \frac{\Psi'(\mu)\mu^t d\mu}{(\Psi(\mu) - \lambda)^2} \tag{139}$$

586 implying

$$|U(t, \lambda)| \leq \frac{1}{2\pi t} \int_{-\pi}^{\pi} \frac{|\Psi'(e^{is})|ds}{|\Psi(e^{is}) - \lambda|^2}. \tag{140}$$

587 We will show that this integral is $O(\frac{1}{\lambda})$.

Note first that we can replace the integration on $[-\pi, \pi]$ by integration on $[-a, a]$ for any $0 < a < \pi$. Indeed, by our assumptions $\Psi$ is $C^1$ on the unit circle, and $\Psi(\mu) = 1$ there only if $\mu = 1$. Accordingly, the remaining part of the integral is non-singular as $\lambda \searrow 0$ and so is uniformly bounded for all $\lambda \in (0, \lambda_{\max}]$.

Recall that by our assumption $\Psi'(\mu) = O(|\mu - 1|^{\theta-1})$ as $\mu \to 1$. Applying again Lemma 3,

$$|U(t, \lambda)| \leq \frac{C}{t} \int_0^\infty \frac{s^{\theta-1} ds}{(s^\theta + \lambda)^2} = \frac{C'}{t\lambda} \tag{141}$$

with some constant $C'$ independent of $t, \lambda$. It follows that

$$|F_U(t\lambda^{1/\theta}, \lambda)| = |\lambda^{1-1/\theta} U(t, \lambda)| \leq \frac{C'}{t\lambda^{1/\theta}}, \tag{142}$$

as claimed. $\qquad\qquad\square$

We return now to proving that replacing $F_U(t\lambda_k^{1/\theta}, \lambda_k)$ by $F_U(t\lambda_k^{1/\theta})$ in Eq. (105) amounts to a lower-order correction $o(t^{\theta/\nu-2})$. It suffices to prove that $\Delta a_t \to 0$, where

$$\Delta a_t = t^{2-\theta/\nu} \sum_k \lambda_k^{2/\theta} (F_U^2(t\lambda_k^{1/\theta}, \lambda_k) - F_U^2(t\lambda_k^{1/\theta})) \tag{143}$$

$$= t^{-\theta/\nu} \sum_k (t\lambda_k^{1/\theta})^2 (F_U^2(t\lambda_k^{1/\theta}, \lambda_k) - F_U^2(t\lambda_k^{1/\theta})). \tag{144}$$

For any interval $I \subset \mathbb{R}$, denote by $\Delta S_{I,t}$ the part of $\Delta a_t$ corresponding to the terms in (144) such that $t\lambda_k^{1/\theta} \in I$. By statement 1 of Lemma 4, for any $u > 0$ we have, as $t \to \infty$,

$$|\Delta S_{(0,u),t}| = o(1) t^{2-\theta/\nu} \sum_{k: t\lambda_k^{1/\theta} < u} \lambda_k^{2/\theta} \tag{145}$$

$$= o(1) t^{2-\theta/\nu} O((t/u)^{(\theta/\nu)(1-2\nu/\theta)}) \tag{146}$$

$$= o(1), \tag{147}$$

where we have used the fact that $2\nu/\theta > \nu > 1$.

Now consider the remaining interval $I = [u, +\infty)$. It suffices to prove that $|\Delta S_{[u,+\infty),t}|$ can be made arbitrarily small uniformly in $t$ by choosing $u$ large enough. By statement 2 of Lemma 4, we can write

$$|\Delta S_{[u,+\infty),t}| \leq C t^{2-\theta/\nu} \sum_{k: t\lambda_k^{1/\theta} > u} \lambda_k^{2/\theta} (t\lambda_k^{1/\theta})^{-2} \tag{148}$$

$$\leq C t^{-\theta/\nu} \sum_{k=1}^{\Lambda_+^{1/\nu}(t/u)^{\theta/\nu}} 1 \tag{149}$$

$$\leq C' u^{-\theta/\nu} \tag{150}$$

with some $t, u$-independent constant $C'$. This completes the proof of statement 1 of Theorem 3.

## C.2  The signal propagators

The proof for the signal propagators follows the same ideas as for the noise propagators, with appropriate adjustments.

**The function $F_V$.** We introduce the values

$$V(t, \lambda) = \frac{1}{2\pi i} \oint_{|\mu|=1} \frac{\Psi(\mu)\mu^{t-1} d\mu}{(\Psi(\mu) - \lambda)(\mu - 1)} = \frac{1}{2\pi} \int_{-\pi}^{\pi} \frac{\Psi(e^{i\phi}) e^{it\phi} d\phi}{(\Psi(e^{i\phi}) - \lambda)(e^{i\phi} - 1)} \tag{151}$$

so that, by Eq. (21), the propagators $V_t$ can be written as

$$V_t = \sum_{k=1}^{\infty} \lambda_k (\mathbf{e}_k^T \mathbf{w}_*)^2 |V(t, \lambda_k)|^2. \tag{152}$$

With the change of variables $\phi = s\lambda^{1/\theta}$,

$$V(t,\lambda) = \frac{\lambda^{1/\theta}}{2\pi} \int_{-\pi/\lambda^{1/\theta}}^{\pi/\lambda^{1/\theta}} \frac{(-\Psi(e^{is\lambda^{1/\theta}})/\lambda)e^{it\lambda^{1/\theta}s}ds}{(-\Psi(e^{is\lambda^{1/\theta}})/\lambda + 1)(e^{is\lambda^{1/\theta}} - 1)} = F_V(t\lambda^{1/\theta}, \lambda), \tag{153}$$

where

$$F_V(r,\lambda) = \frac{\lambda^{1/\theta}}{2\pi} \int_{-\pi/\lambda^{1/\theta}}^{\pi/\lambda^{1/\theta}} \frac{(-\Psi(e^{is\lambda^{1/\theta}})/\lambda)e^{irs}ds}{(-\Psi(e^{is\lambda^{1/\theta}})/\lambda + 1)(e^{is\lambda^{1/\theta}} - 1)}. \tag{154}$$

We again recall that $\Psi(\mu) = -c_\Psi(\mu-1)^\theta(1+o(1))$ as $\mu \to 1$ and formally take the pointwise limit $\lambda \searrow 0$ in the integrand to obtain the expression

$$F_V(r,0) \overset{\text{def}}{=} F_V(r) \overset{\text{def}}{=} \frac{1}{2\pi i} \int_{-\infty}^{\infty} \frac{c_\Psi e^{i(\text{sign } s)\theta\pi/2}|s|^\theta e^{irs}ds}{(c_\Psi e^{i(\text{sign } s)\theta\pi/2}|s|^\theta + 1)s} \tag{155}$$

$$= \frac{1}{2\pi} \int_{-\infty}^{\infty} \frac{c_\Psi e^{i(\text{sign } s)(\theta-1)\pi/2}|s|^{\theta-1}e^{irs}ds}{(c_\Psi e^{i(\text{sign } s)\theta\pi/2}|s|^\theta + 1)} \tag{156}$$

for any fixed $r$. This integral can be equivalently written as

$$F_V(r) = \frac{1}{2\pi i} \int_{i\mathbb{R}} \frac{c_\Psi z^{\theta-1}e^{rz}dz}{c_\Psi z^\theta + 1}, \tag{157}$$

assuming again the standard branch of $z^\theta$ holomorphic in $\mathbb{C} \setminus (-\infty, 0]$. The function $F_V$ can be written in terms of the Mittag-Leffler function $E_\theta \equiv E_{\theta,1}$ (the special case of $E_{a,b}$ given by Eq. (87)):

$$F_V(r) = E_\theta\left(-\frac{r^\theta}{c_\Psi}\right). \tag{158}$$

Note that, in contrast to $F_U$, the integrals (156), (157) are not absolutely summable, due to the $z^{-1}$ fall off of the integrand at $z \to \infty$. However, the integrand is square-summable and so $F_V$, as a Fourier transform of such function, is well-defined almost everywhere as a square-integrable function.

In fact, $F_V$ can be defined for each particular $r \neq 0$ by restricting the integration in (156) to segments $[u, v]$ and letting $u \to -\infty$ and $v \to \infty$. Indeed, the resulting Fourier transforms $F_V^{(u,v)}$ converge to $F_V$ in $L^2(\mathbb{R})$. However, these transforms are continuous functions of $r$, and as $u \to \infty, v \to \infty$ they converge pointwise, and even uniformly on the sets $\{r : |r| > \epsilon\}$, for any fixed $\epsilon > 0$.

To see this last property of uniform pointwise convergence, note that the integrand in (156) has the form $(s^{-1} + O(s^{-1-\theta}))e^{irs}$ as $s \to \infty$. The component $O(s^{-1-\theta}))$ is in $L^1$, so the respective part of $F_V^{(u,v)}$ converges as $u \to -\infty, v \to \infty$ uniformly for all $r \in \mathbb{R}$. Regarding the $s^{-1}$ component, integrating by parts gives

$$\int_1^v \frac{e^{irs}ds}{s} = \frac{e^{irs}}{irs}\Big|_{s=1}^v + \frac{1}{ir}\int_1^v \frac{e^{irs}ds}{s^2}. \tag{159}$$

This expression converges as $v \to \infty$ uniformly for $\{r : |r| > \epsilon\}$ with any fixed $\epsilon > 0$, as claimed. The same argument applies to $\int_u^{-1}$.

The above argument shows, in particular, that $F_V$ is naturally defined as a function continuous on the intervals $(0, +\infty)$ and $(-\infty, 0)$.

We collect further properties of $F_V(r)$ in the following lemma that parallels Lemma 1 for $F_U$. The proofs are also similar to the proofs in Lemma 1.

**Lemma 5.**

    *1. $F_V(r) = 0$ for $r < 0$.*

    *2. $F_V(r) \to 1$ as $r \searrow 0$.*

    *3. $F_V(r) = (1 + o(1))\frac{c_\Psi}{\Gamma(1-\theta)}r^{-\theta}$ as $r \to +\infty$.*

*Proof.* 1. Like in Lemma 1, this follows by deforming the integration line in Eq. (157) towards $+\infty$.

2. By the change of variables $rz = z'$,

$$F_V(r) = \frac{1}{2\pi i} \int_{i\mathbb{R}} \frac{z'^{\theta-1} e^{z'} dz'}{z'^{\theta} + c_\Psi^{-1} r^{\theta}}. \tag{160}$$

As in Lemma 1, the integration line $i\mathbb{R}$ can be deformed to the line $\gamma_a, a > 0$, encircling the negative semi-axis:

$$\gamma_a = \gamma_{a,1} \cup \gamma_{a,2} \cup \gamma_{a,3}, \tag{161}$$
$$\gamma_{a,1} = \{z \in \mathbb{C} : \Im z = -a, \Re z \leq 0\}, \tag{162}$$
$$\gamma_{2,2} = \{z \in \mathbb{C} : |z| = a, -\tfrac{\pi}{2} < \arg z < \tfrac{\pi}{2})\}, \tag{163}$$
$$\gamma_{a,1} = \{z \in \mathbb{C} : \Im z = a, \Re z \leq 0\}. \tag{164}$$

Taking the limit $r \searrow 0$, we get

$$\lim_{r \searrow 0} F_V(r) = \lim_{r \searrow 0} \frac{1}{2\pi i} \int_{\gamma_a} \frac{z'^{\theta-1} e^{z'} dz'}{z'^{\theta} + c_\Psi^{-1} r^{\theta}} = \frac{1}{2\pi i} \int_{\gamma_a} \frac{e^{z'} dz'}{z'} = 1, \tag{165}$$

since the last integral simply amounts to the residue of $e^{z'}/z'$ at $z' = 0$.

3. Using the same contour $\gamma'$ as in Lemma 1,

$$F_V(r) = v(r) r^{-\theta}, \quad v(r) = \frac{1}{2\pi i} \int_{\gamma'} \frac{c_\Psi z'^{\theta-1} e^{z'} dz'}{c_\Psi (z'/r)^{\theta} + 1}. \tag{166}$$

Taking the limit $r \to +\infty$ and deforming the contour to the negative semi-axis as in Lemma 1,

$$\lim_{r \to +\infty} v(r) = \frac{c_\Psi}{2\pi i} \int_{\gamma'} z'^{\theta-1} e^{z'} dz' = \frac{c_\Psi}{\Gamma(1-\theta)}. \tag{167}$$

$\square$

**The formal leading term in $V_t$.** We have

$$V_t = \sum_{k=1}^{\infty} \lambda_k (\mathbf{e}_k^T \mathbf{w}_*)^2 |V(t, \lambda_k)|^2 = \sum_k \lambda_k (\mathbf{e}_k^T \mathbf{w}_*)^2 F_V^2(t\lambda_k^{1/\theta}, \lambda_k). \tag{168}$$

To extract the leading term in this expression, we set the second argument in $F_V(t\lambda_k^{1/\theta}, \lambda_k)$ to 0:

$$V_t^{(1)} \stackrel{\text{def}}{=} \sum_k \lambda_k (\mathbf{e}_k^T \mathbf{w}_*)^2 F_V^2(t\lambda_k^{1/\theta}) = b_t t^{-\theta\zeta}, \tag{169}$$

where

$$b_t = t^{\theta\zeta} \sum_k \lambda_k (\mathbf{e}_k^T \mathbf{w}_*)^2 F_V^2(t\lambda_k^{1/\theta}). \tag{170}$$

The analog of Lemma 2 is

**Lemma 6.**

$$\lim_{t \to \infty} b_t = Q \int_0^\infty F_V^2(r) dr^{\theta\zeta} = Q\theta\zeta \int_0^\infty r^{\theta\zeta-1} F_V^2(r) dr < \infty. \tag{171}$$

*Proof.* First, observe that, by the source condition (12) and Lemma 5, the integral converges near $r = 0$ since $\theta\zeta > 0$, and near $r = \infty$ since $\zeta < 2$.

We can establish convergence of the sequence $b_t$ using the same steps as in Lemma 2. We first introduce the sums $S_{I,t}$ comprising the terms of expansion (170) such that $t\lambda_k^{1/\theta} \in I$. For intervals $I = [u, v]$ with $0 < u < v < \infty$ we show, using the source condition (12) and approximation by integral sums, that

$$\lim_{t \to \infty} S_{I,t} = t^{\theta\zeta} \int_I F_V^2(r) dQ((r/t)^{\theta})^{\zeta} = Q \int_I F_V^2(r) dr^{\theta\zeta}. \tag{172}$$

After that we show that the contribution of the remaining intervals $(v, +\infty)$ and $(0, u)$ can be made arbitrarily small uniformly in $t$ by adjusting $u, v$.

In particular, consider the interval $I = (v, +\infty)$. Let $R(\lambda) = \sum_{k:\lambda_k \leq \lambda} \lambda_k (\mathbf{e}_k^T \mathbf{w}_*)^2$ denote the cumulative distribution function of the spectral measure. Since the spectral measure is compactly supported, assumption (12) implies that $R(\lambda) \leq Q' \lambda^\zeta$ for all $\lambda > 0$ with some $Q' > 0$. Using statement 3 of Lemma 5 and integration by parts, we can bound

$$S_{(v,+\infty),t} \leq t^{\theta\zeta} \sum_{k:t\lambda_k^{1/\theta} > v} \lambda_k (\mathbf{e}_k^T \mathbf{w}_*)^2 C(t\lambda_k^{1/\theta})^{-2\theta} \tag{173}$$

$$= Ct^{\theta(\zeta-2)} \int_{(v/t)^\theta}^\infty \frac{dR(\lambda)}{\lambda^2} \tag{174}$$

$$= Ct^{\theta(\zeta-2)} \left( \frac{R(\lambda)}{\lambda^2} \Big|_{(v/t)^\theta}^\infty + 2 \int_{(v/t)^\theta}^\infty \frac{R(\lambda)d\lambda}{\lambda^3} \right) \tag{175}$$

$$\leq 2CQ' t^{\theta(\zeta-2)} \int_{(v/t)^\theta}^\infty \lambda^{\zeta-3} d\lambda \tag{176}$$

$$\leq C' v^{(\zeta-2)\theta} \tag{177}$$

with some constant $C'$ independent of $v, t$.

For the intervals $I = (0, u)$ we have

$$S_{(0,u),t} \leq t^{\theta\zeta} \sum_{k:t\lambda_k^{1/\theta} < u} \lambda_k (\mathbf{e}_k^T \mathbf{w}_*)^2 C \tag{178}$$

$$\leq Ct^{\theta\zeta} Q((u/t)^\theta)^\zeta \tag{179}$$

$$= C' u^{\theta\zeta}. \tag{180}$$

$\square$

**Completion of proof.** It remains to show that the correction in $V_t$ due to the replacement of $F_V(t\lambda_k^{1/\theta}, \lambda_k)$ by $F_V(t\lambda_k^{1/\theta})$ in Eq. (168) is $o(t^{-\theta\zeta})$. We first establish an analog of Lemma 4:

**Lemma 7.** *Assuming that $r = t\lambda^{1/\theta}$ with $t = 1, 2, \ldots$:*

   *1. $|F_V(r, \lambda) - F_V(r)| = o(1)$ as $\lambda \to 0$, uniformly for $r > \epsilon$, for any $\epsilon > 0$.*

   *2. $|F_V(r, \lambda)| \leq C \min(\frac{1}{r}, 1)$ for all $t = 1, 2, \ldots$ and $\lambda \in (0, \lambda_{\max}]$, with some $r, \lambda$-independent constant $C$.*

*Proof.* 1. The proof of this property is more complicated than the earlier proof for $F_U$ because the integrals defining $F_V$ are not absolutely convergent. Recall the integration by parts argument (159) used to define $F_V(r)$ as the pointwise limit of the functions $F_V^{(u,v)}(r)$. We extend this approach to the functions $F_V(r, \lambda)$ with $\lambda > 0$. Specifically, let $F_V^{(u)}(r, \lambda)$ be defined as $F_V(r, \lambda)$ in Eq. (154), but with integration restricted to the segment $[-u, u]$. By analogy with our convention $F_V(r) \equiv F_V(r, \lambda = 0)$, denote also $F_V^{(u)}(r) \equiv F_V^{(u)}(r, \lambda = 0)$. We will establish the following two properties:

   (a) $|F_V^{(u)}(r, \lambda) - F_V(r, \lambda)| \leq \frac{C}{ru}$ for all $0 < \lambda < \lambda_{\max}$ with a $r, u, \lambda$-independent constant $C$.

   (b) For any $u$, $|F_V^{(u)}(r, \lambda) - F_V^{(u)}(r)| \to 0$ as $\lambda \searrow 0$ uniformly for $r \in \mathbb{R}$.

Observe first that these two properties imply the claimed uniform convergence $|F_V(r, \lambda) - F_V(r)| = o(1)$ as $\lambda \to 0$. Indeed, given any $\delta > 0$, first set $u = \frac{3C}{\epsilon}$ so that by (a) we have

$$|F_V^{(u)}(r, \lambda) - F_V(r, \lambda)| \leq \delta/3 \tag{181}$$

for all $r > \epsilon$ and $0 < \lambda < \lambda_{\max}$. This inequality also holds in the limit $\lambda \searrow 0$, i.e.

$$|F_V^{(u)}(r) - F_V(r)| \le \delta/3. \tag{182}$$

Now (b) implies that for sufficiently small $\lambda$ we have

$$|F_V^{(u)}(r, \lambda) - F_V^{(u)}(r)| \le \delta/3 \tag{183}$$

uniformly in $r \in \mathbb{R}$. Combining all three above inequalities, we see that for sufficiently small $\lambda$

$$|F_V(r, \lambda) - F_V(r)| \le \delta \tag{184}$$

uniformly for $r > \epsilon$, as desired.

It remains to prove the statements (a) and (b). Statement (b) immediately follows from the uniform $\lambda \searrow 0$ convergence of the integrand in expression (154) on the interval $s \in [-u, u]$.

To prove statement (a), we perform integration by parts, using the $\frac{2\pi}{\lambda^{1/\theta}}$-periodicity of the integrand:

$$|F_V^{(u)}(r, \lambda) - F_V(r, \lambda)| \tag{185}$$

$$= \frac{\lambda^{1/\theta}}{2\pi} \left| \int_{[-\frac{\pi}{\lambda^{1/\theta}}, \frac{\pi}{\lambda^{1/\theta}}] \setminus [-u,u]} \frac{(\Psi(e^{is\lambda^{1/\theta}})/\lambda) e^{irs} ds}{(\Psi(e^{is\lambda^{1/\theta}})/\lambda - 1)(e^{is\lambda^{1/\theta}} - 1)} \right| \tag{186}$$

$$= \frac{\lambda^{1/\theta}}{2\pi r} \left| \frac{(\Psi(e^{is\lambda^{1/\theta}})/\lambda) e^{irs}}{(\Psi(e^{is\lambda^{1/\theta}})/\lambda - 1)(e^{is\lambda^{1/\theta}} - 1)} \right|_{s=u}^{-u} - \int_{[-\frac{\pi}{\lambda^{1/\theta}}, \frac{\pi}{\lambda^{1/\theta}}] \setminus [-u,u]} \tag{187}$$

$$\frac{i\lambda^{1/\theta}[(-\Psi'(e^{is\lambda^{1/\theta}})/\lambda)(e^{is\lambda^{1/\theta}} - 1) - (\Psi(e^{is\lambda^{1/\theta}})/\lambda)(\Psi(e^{is\lambda^{1/\theta}})/\lambda - 1)e^{is\lambda^{1/\theta}}] e^{irs} ds}{(\Psi(e^{is\lambda^{1/\theta}})/\lambda - 1)^2 (e^{is\lambda^{1/\theta}} - 1)^2} \Bigg|.$$

By our assumptions on $\Psi$, Lemma 3 and standard inequalities, there exist $\lambda, s$-independent constants $C, c > 0$ such that for all $\lambda \in (0, \lambda_{\max}]$ and $s \in [-\frac{\pi}{\lambda^{1/\theta}}, \frac{\pi}{\lambda^{1/\theta}}]$

$$|\Psi(e^{is\lambda^{1/\theta}})| \le C|s|^\theta \lambda, \tag{188}$$

$$|\Psi'(e^{is\lambda^{1/\theta}})| \le C\theta |s|^{\theta-1} \lambda^{(\theta-1)/\theta}, \tag{189}$$

$$|\Psi(e^{is\lambda^{1/\theta}})/\lambda - 1| \ge c(1 + |s|^\theta), \tag{190}$$

$$|e^{is\lambda^{1/\theta}} - 1| \ge c|s|\lambda^{1/\theta}. \tag{191}$$

Applying these inequalities to Eq. (187), we find that

$$|F_V^{(u)}(r, \lambda) - F_V(r, \lambda)| \le \frac{C'}{r} \left( \frac{u^\theta}{(1 + u^\theta)u} + \int_{[-\frac{\pi}{\lambda^{1/\theta}}, \frac{\pi}{\lambda^{1/\theta}}] \setminus [-u,u]} \frac{|s|^\theta ds}{(1 + |s|^\theta)s^2} \right) \tag{192}$$

$$\le \frac{C''}{ru}, \tag{193}$$

as desired.

2. Note that

$$|F_V(r, \lambda)| \le \frac{C}{r}, \quad C < \infty, \tag{194}$$

simply by setting $u = 0$ in the bound (192), since the first term on the r.h.s. of (192) vanishes and the second converges thanks to $\theta > 1$.

It remains to prove that $F_V(r, \lambda)$ is bounded uniformly in $r, \lambda$. It suffices to prove this for $r < \epsilon$ with some fixed $\epsilon > 0$, since for larger $r$ this follows from bound (194). Since $r = t\lambda^{1/\theta}$, this means it is sufficient to consider

$$\lambda \le (\epsilon/t)^\theta. \tag{195}$$

To this end consider the original definition (151) of $V(t, \lambda)$ in terms of integration over the contour $\{|\mu| = 1\}$. We will deform this contour within the analiticity domain $\{\mu \in \mathbb{C} : |\mu| \ge 1\}$ to another contour $\gamma$, to be specified below, that fully encircles the point $\mu = 1$:

$$V(t, \lambda) = \frac{1}{2\pi i} \oint_\gamma \frac{\Psi(\mu)\mu^{t-1} d\mu}{(\Psi(\mu) - \lambda)(\mu - 1)}. \tag{196}$$

It is convenient to subtract the residue of $\mu^{t-1}/(\mu-1)$ equal to 1:

$$V(t,\lambda) - 1 = \frac{1}{2\pi i}\oint_\gamma \frac{\Psi(\mu)\mu^{t-1}d\mu}{(\Psi(\mu)-\lambda)(\mu-1)} - \frac{1}{2\pi i}\oint_\gamma \frac{\mu^{t-1}d\mu}{\mu-1} = \frac{\lambda}{2\pi i}\oint_\gamma \frac{\mu^{t-1}d\mu}{(\Psi(\mu)-\lambda)(\mu-1)}.$$
(197)

We define now $\gamma$ as the original contour perturbed to include an arc of radius $1/t$ centered at 1:

$$\gamma = \gamma_1 \cup \gamma_2,$$
(198)

$$\gamma_1 = \{e^{i\phi}\}_{\phi_1 \le \phi \le 2\pi - \phi_1},$$
(199)

$$\gamma_2 = \{1 + \tfrac{e^{i\phi}}{t}\}_{-\phi_2 \le \phi \le \phi_2},$$
(200)

where $\phi_1 \in (0, \frac{\pi}{2})$, $\phi_2 \in (\frac{\pi}{2}, \pi)$ are such that $\gamma$ is connected. Note that $\phi_1 \propto \frac{1}{t}$ as $t \to \infty$.

Now we bound separately the contribution to the integral from $\gamma_1$ and $\gamma_2$. For $\gamma_1$ and $-\pi \le \phi \le \pi$ we use the inequalities

$$|\Psi(e^{i\phi}) - \lambda| \ge c|\phi|^\theta,$$
(201)

$$|e^{i\phi} - 1| \ge c|\phi|$$
(202)

with a $\phi, \lambda$-independent constant $c > 0$. This gives, using Eq. (195),

$$\lambda\left|\int_{\gamma_1} \frac{\mu^{t-1}d\mu}{(\Psi(\mu)-\lambda)(\mu-1)}\right| \le \lambda C\left|\int_{-\pi}^{-\phi_1} + \int_{\phi_1}^{\pi} \frac{d\phi}{|\phi|^{\theta+1}}\right| \le C'\frac{\lambda}{\phi_1^\theta} \le C''\lambda t^\theta \le C''\epsilon^\theta.$$
(203)

For the $\gamma_2$ component we use the inequalities

$$|1 + \tfrac{e^{i\phi}}{t}|^{t-1} \le e,$$
(204)

$$|\Psi(1 + \tfrac{e^{i\phi}}{t}) - \lambda| \ge ct^{-\theta}, \quad -\phi_2 \le \phi \le \phi_2.$$
(205)

(Inequality (205) relies on the assumption $\theta < 2$ and can be proved similarly to Lemma 3.) This gives

$$\lambda\left|\int_{\gamma_2} \frac{\mu^{t-1}d\mu}{(\Psi(\mu)-\lambda)(\mu-1)}\right| \le \lambda C\left|\int_{-\pi}^{\pi} \frac{t^{-1}d\phi}{t^{-\theta}\cdot t^{-1}}\right| \le C'\lambda t^\theta \le C''\epsilon^\theta.$$
(206)

Fixing some $\epsilon > 0$, we see from Eqs. (203), (206) that under assumption (195) the expressions $|V(t,\lambda) - 1|$, and hence $|V(t,\lambda)|$, are uniformly bounded, as desired.

This completes the proof of the lemma. $\qquad\square$

This lemma can now be used to show that replacing $F_V(t\lambda_k^{1/\theta}, \lambda_k)$ by $F_V(t\lambda_k^{1/\theta})$ in Eq. (168) amounts to a lower-order correction $o(t^{-\theta\zeta})$ in the propagator $V_t$. The argument is similar to the respective argument for $F_U$ in the end of Section C.1. Statement 1 of Lemma 7 is used to show this for the contribution of the terms $k$ with $u < t\lambda_k^{1/\theta} < v$, for any $0 < u < v < +\infty$. Then, for terms with $t\lambda_k^{1/\theta} < u$ we use the uniform boundedness of $F_V(r,\lambda)$, i.e. the part $F_V(r,\lambda) \le C$ of statement 2, and show that their contribution can be made arbitrarily small by decreasing $u$. Finally, for terms with $t\lambda_k^{1/\theta} > v$ we use the part $F_V(r,\lambda) \le \frac{C}{r}$ of statement 2, and show that their contribution can be made arbitrarily small by increasing $v$.

This completes the proof of Theorem 3.

## D Proof of Proposition 1

To simplify notation, set $A = 1$; results for general $A$'s are easily obtained by rescaling.

Note first that for any $\mu \in \mathbb{C} \setminus [0, 1]$ the integral in Eq. (28) converges and is nonzero. To see that it is nonzero, note that if $\mu$ has a nonzero imaginary part, then the integral has a nonzero imaginary part of the opposite sign, hence is nonzero. On the other hand, if $\mu > 1$ or $\mu < 0$, then the integral is strictly positive or negative, so also nonzero. It follows that the expression in parentheses is invertible and so $\Psi(\mu)$ is well-defined for all $\mu \in \mathbb{C} \setminus [0, 1]$.

734 The asymptotics $\Psi(\mu) = -\mu(1 + o(1))$ at $\mu \to \infty$ is obvious.

735 To find the asymptotics at $\mu \to 1$, make the substitution $z = \delta/(\mu - 1)$ in the integral:

$$\int_0^1 \frac{d\delta^{2-\theta}}{\mu - 1 + \delta} = (\mu - 1)^{\theta-1} \int_0^{1/(\mu-1)} \frac{dz^{2-\theta}}{1 + z}. \tag{207}$$

736 As $\mu \to 1$ the last integral converges to a standard integral:

$$\int_0^{1/(\mu-1)} \frac{dz^{2-\theta}}{1 + z} \to \int_0^\infty \frac{dz^{2-\theta}}{1 + z} = \frac{(2 - \theta)\pi}{\sin((2 - \theta)\pi)}. \tag{208}$$

737 The integration line in the last integral is any line connecting 0 to $\infty$ in $\mathbb{C} \setminus (-\infty, 0)$; the integral
738 does not depend on the line thanks to the condition $\theta > 1$.

739 We prove now that $\Psi(\{|\mu| \geq 1\}) \cap (0, 2] = \varnothing$. Let us first show that if $|\mu| \geq 1$ and $\Im\mu \neq 0$, then
740 $\Psi(\mu) \notin (0, +\infty)$. To this end write

$$\Psi(\mu) = ab, \tag{209}$$

$$a = -\Big(\int_0^1 \frac{(\mu - 1)d\delta^{2-\theta}}{\mu - 1 + \delta}\Big)^{-1} = -\Big(\int_0^1 \frac{d\delta^{2-\theta}}{1 + \frac{\delta}{\mu-1}}\Big)^{-1}, \tag{210}$$

$$b = \frac{(\mu - 1)^2}{\mu} = J(\mu) - 2, \tag{211}$$

741 where $J(\mu) = \mu + \frac{1}{\mu}$ is Zhukovsky's function.

742 Suppose, for definiteness, that $\Im\mu > 0$. Regarding $a$, note that if $\Im\mu > 0$, then the imaginary part
743 of the integrand in Eq. (210) is also positive, and so $\Im a > 0$.

744 Regarding $b$, recall that if $\Im\mu > 0$ and $|\mu| > 1$, then $\Im J(\mu) > 0$. On the other hand, if $|\mu| = 1$,
745 then $J(\mu) \in [-2, 2]$. Combining these observations, we see that if $\Im\mu > 0$ and $|\mu| \geq 1$, then either
746 $\Im b > 0$, or $b \leq 0$. Since $\Im a > 0$, it follows that $ab \notin (0, +\infty)$.

747 We see that $\Psi(\mu)$ can be real and positive only if $\mu \in \mathbb{R}$. Clearly, $\Psi(\mu) > 0$ if $\mu \leq -1$, and
748 $\Psi(\mu) \leq 0$ if $\mu \geq 1$. It is easily checked by differentiation that $\Psi(\mu)$ is monotone decreasing for
749 $\mu \in (-\infty, -1]$, so the smallest positive value attained by $\Psi$ is

$$\Psi(-1) = 2\Big(\int_0^1 \frac{d\delta^{2-\theta}}{2 - \delta}\Big)^{-1} > 2. \tag{212}$$

## 750 E   Proof of Proposition 2

751 In terms of $\alpha, \mathbf{b}, \mathbf{c}, D$, the components $P, Q$ of the characteristic polynomial $\det(\mu - S_\lambda) = P(\mu) - $
752 $\lambda Q(\mu)$ can be written as

$$P(\mu) = (\mu - 1)\det(\mu - D), \tag{213}$$

$$Q(\mu) = -\det\begin{pmatrix} \alpha & \mathbf{b}^T \\ \mathbf{c} & \mu - D \end{pmatrix} = \det(\mu - D)(\mathbf{b}^T(\mu - D)^{-1}\mathbf{c} - \alpha). \tag{214}$$

753 (see Theorem 1 in [29]). Accordingly,

$$\frac{(\mu - 1)Q(\mu)}{P(\mu)} = \mathbf{b}^T(\mu - D)^{-1}\mathbf{c} - \alpha. \tag{215}$$

754 If $D = \mathrm{diag}(d_1, \ldots, d_M)$, then

$$\frac{(\mu - 1)Q(\mu)}{P(\mu)} = \sum_{m=1}^M \frac{b_m c_m}{\mu - d_m} - \alpha. \tag{216}$$

On the other hand, our definition of $\Psi^{(M)}$ implies that

$$\frac{(\mu-1)A}{\Psi^{(M)}(\mu)} = (\theta-2)h\mu\sum_{m=1}^{M}\frac{e^{-(2-\theta)(m-1/2)h}}{\mu-1+e^{-(m-1/2)h}} \tag{217}$$

$$= (\theta-2)h\Big[\sum_{m=1}^{M}\frac{e^{-(2-\theta)(m-1/2)h}(1-e^{-(m-1/2)h})}{\mu-1+e^{-(m-1/2)h}} + \sum_{m=1}^{M}e^{-(2-\theta)(m-1/2)h}\Big] \tag{218}$$

$$= (2-\theta)h\Big[\sum_{m=1}^{M}\frac{e^{-(2-\theta)(m-1/2)h}(e^{-(m-1/2)h}-1)}{\mu-1+e^{-(m-1/2)h}} - \frac{1-e^{-(2-\theta)Mh}}{1-e^{-(2-\theta)h}}e^{-(2-\theta)h/2}\Big]. \tag{219}$$

By comparing this expansion with Eq. (216), we see that the values of $\alpha, \mathbf{b}, \mathbf{c}, D$ given in Eqs. (31)-(34) ensure that $P/Q = \Psi^{(M)}$.

## F  Experiments

The experiments in this section[2] are performed with Corner SGD approximated as in Proposition 2 with memory size $M = 5$ and spacing parameter $l = 5$. Experiments have been performed with GPU NVIDIA GeForce RTX 4070, CPU Intel Core i5-12400F, and 32 GB RAM; the training of all the models on GPU has taken less than half an hour.

**A synthetic indicator problem.**  Suppose that we are fitting the indicator function $y(x) = \mathbf{1}_{[1/4,3/4]}(x)$ on the segment $[0,1]$ using the shallow ReLU neural network in which only the output layer weights $w_n$ are trained:

$$\widehat{y}(x, \mathbf{w}) = \frac{1}{N}\sum_{n=1}^{N}w_n(x-\tfrac{n}{N})_+, \quad (x)_+ \equiv \max(x,0). \tag{220}$$

This is an exactly linear model that in the limit $N \to \infty$ acquires the form

$$\widehat{y}(x) = \int_0^1 w(y)(x-y)_+dy = \mathbf{x}^T\mathbf{w}, \tag{221}$$

where $\mathbf{x}, \mathbf{w}$ are understood as vectors in $L^2([0,1])$, and $\mathbf{x} \equiv (x-\cdot)_+$. We consider the loss $L(\mathbf{w}) = \mathbb{E}_{x\sim U(0,1)}\frac{1}{2}(\mathbf{x}^T\mathbf{w} - y(x))^2$, where $U(0,1)$ is the uniform distribution on $[0,1]$.

This limiting integral problem obeys asymptotic spectral power laws (11),(12) with precisely computable $\nu, \zeta$ (see Appendix H):

$$\zeta = \tfrac{1}{4}, \quad \nu = 4. \tag{222}$$

The problem thus falls into the sub-phase I "full acceleration" of the signal dominated phase, and we expect that it can be accelerated with corner algorithms up to $\theta_{\max} = 2$.

In the experiment we set $N = 10^5$ and apply corner SGD with $\theta = 1.8$, see Figure 3. The experimental exponent of plain SGD is close to the theoretical value $\zeta = 0.25$. The accelerated exponent of approximate Corner SGD is slightly lower, but close to the theoretical value $\theta\zeta = 1.8 \cdot 0.25 = 0.45$.

**MNIST.**  We consider MNIST [15] digit classification performed by a single-hidden-layer ReLU neural network:

$$\widehat{y}_r(\mathbf{x}, \mathbf{w}) = \frac{1}{\sqrt{H}}\sum_{n=1}^{H}w_{rn}^{(2)}\Big(\sum_{m=1}^{28\times28}w_{nm}^{(1)}x_m\Big)_+, \quad r = 0, \ldots 9. \tag{223}$$

Here, the input vector $\mathbf{x} = (x_m)_{m=1}^{28\times28}$ represents a MNIST image, and the outputs $y_r$ represent the 10 classes. We use the one-hot encoding for the targets $\mathbf{y}(\mathbf{x})$ and the quadratic pointwise loss $\ell(\mathbf{x}, \mathbf{w}) = \frac{1}{2}|\widehat{\mathbf{y}}(\mathbf{x}, \mathbf{w}) - \mathbf{y}(\mathbf{x})|$ for training. The trainable weights include both first- and second-layer weights $w_{nm}^{(1)}, w_{rn}^{(2)}$.

---

[2]A jupyter notebook with all experiments is provided in SM

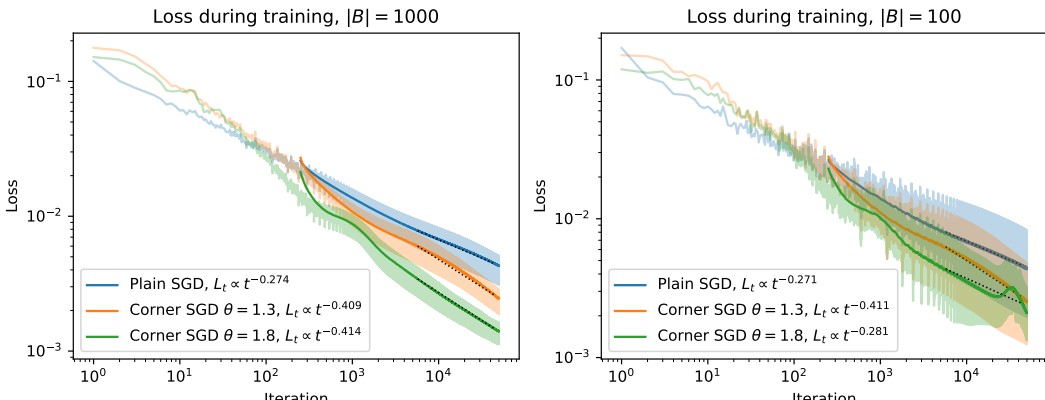

Figure 5: Training loss of neural network (223) on MNIST classification with $H = 1000$, with batch size $|B| = 1000$ (**left**) or 100 (**right**). The full color curves show the smoothed losses.

Note that the model (223) is nonlinear, but for large width $H$ and standard independent weight initialization it belongs to the approximately linear NTK regime [14]. In [26] MNIST was found to have an approximate power-law spectrum with

$$\zeta \approx 0.25, \quad \nu \approx 1.3, \tag{224}$$

putting this problem in the sub-phase III "limited by $U_\Sigma$-finiteness" of the signal-dominated phase (see Figure 1). Theoretically, by Theorem 4, the largest feasible acceleration in this case is $\theta_{\max} = \nu$. Note, however, that this theoretical prediction relied on the infinite-dimensionality of the problem and the divergence of the series $\sum_{t=1}^{\infty} t^{\theta/\nu-2}$. The actual MNIST problem is finite-dimensional, so its $U_\Sigma$ is always finite (though possibly large) and can be made $< 1$ if $|B|$ is large enough. This suggests that corner SGD might practically be used with $\theta > \nu$ and possibly display acceleration beyond the theoretical bound $\theta_{\max} = \nu$. Note also that with exponents (224) the signal/noise balance bound $\frac{2}{\zeta+1/\nu} \approx 2$, i.e. it is not an obstacle for increasing the parameter $\theta$ towards 2.

In Figure 5 we test corner SGD with $\theta = 1.3$ or 1.8 on batch sizes $|B| = 1000$ and 100. The $\theta = 1.3$ version shows a stable performance accelerating the plain SGD exponent $\zeta$ by a factor $\sim 1.5$. The $\theta = 1.8$ version shows lower losses, but does not significantly improve acceleration factor 1.5 at $|B| = 1000$ and is unstable at $|B| = 100$.

In Figure 6 we show both train and test trajectories of the loss and error rate (fraction of incorrectly classified images). The test performance is computed on the standard set of 10000 images, while the training performance is computed by averaging the training loss trajectory. We observe that, similarly to the training set performance, the test performance also improves faster with Corner SGD than with plain SGD. The instability of Corner SGD with $\theta = 1.8$ and batch size 100 observed previously on the training set is also visible on the test set.

## G Additional notes and discussion

**Extension to SE approximation with $\tau_2 \neq 0$.** The key assumption in our derivation and analysis of the contour representation and corner algorithms was the Spectrally Expressible approximation with $\tau_2 = 0$ for the SGD moment evolution (see Eq. (6)). While the SE approximation in general was justified from several points of view in [25, 29], a natural question is how important is the condition $\tau_2 = 0$. This condition substantially simplifies the representation of propagators $U_t, V_t$ in Eqs. (8), but does not seem to correspond to any specific natural data distribution $\rho$. (In contrast, the cases $\tau_1 = \tau_2 = 1$ and $\tau_1 = 1, \tau_2 = -1$ exactly describe translation-invariant and Gaussian distributions; see [25].)

In fact, our analysis of the corner propagators $U_t, V_t$ can be extended from $\tau_2 = 0$ to general $\tau_2$ by a perturbation theory around $\tau_2 = 0$. In Appendix I we sketch an argument suggesting that, at least for sufficiently large batch sizes $|B|$, Theorem 3 remains valid for general $\tau_2$, even with the same coefficients $C_U, C_V$ (i.e., the contribution from $\tau_2 \neq 0$ produces only subleading terms in $U_t, V_t$).

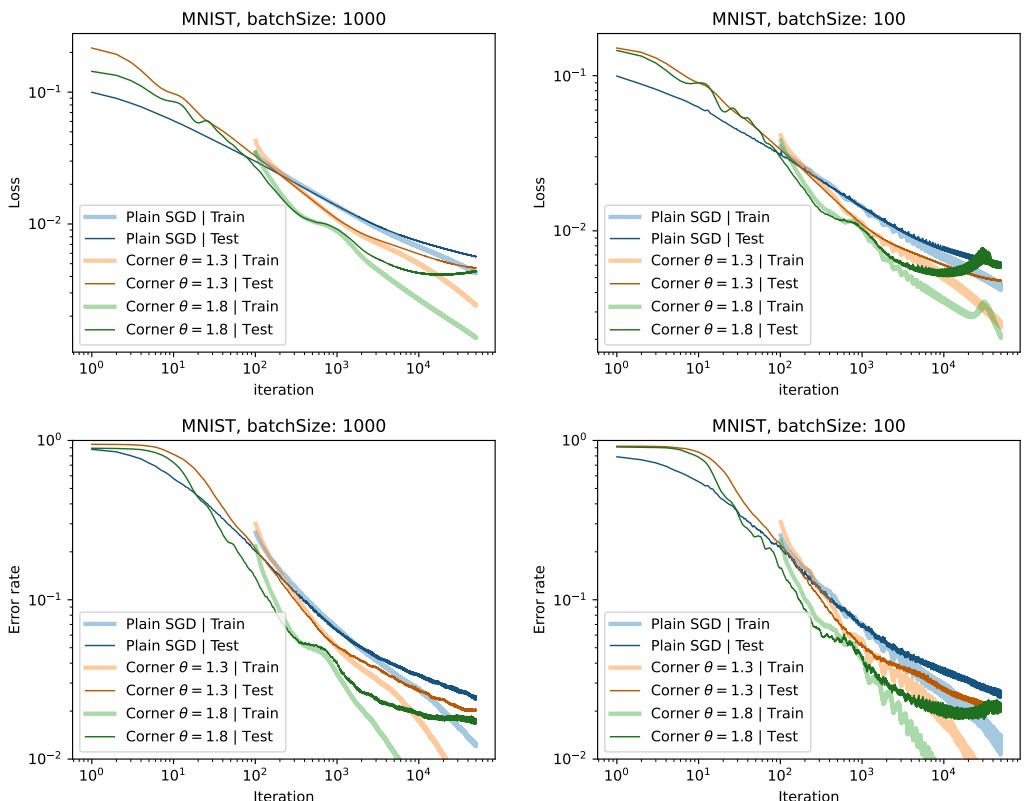

Figure 6: MNIST trajectories of loss **(top row)** and error rate **(bottom row)** on train set **(lighter colors)** and test set **(darker colors)**. **Left column:** batch size 1000. **Right column:** batch size 100.

This implies, in particular, that the acceleration phase diagram in Theorem 4 and Figure 1 (right) is not only $\tau_1$-, but also $\tau_2$-independent.

**Computational complexity.** The main overhead of finitely-approximated corner algorithms compared to plain SGD lies in the memory requirements: if the model has $W$ weights (i.e., $\dim \mathbf{w}_t = W$ in Eq. (1)), then a memory-$M$ algorithm needs to additionally store $MW$ scalars in the auxiliary vectors $\mathbf{u}_t$. On the other hand, the number of elementary operations (arithmetic operations and evaluations of standard elementary functions) in a single iteration of a finitely-approximated corner algorithm need not be much larger than for plain SGD.

Indeed, an iteration (1) of a memory-$M$ algorithm consists in computing the gradient $\nabla L(\mathbf{w}_t)$ and performing a linear transformation. In SGD with batch size $|B|$, the estimated gradient $\nabla L_{B_t}(\mathbf{w}_t)$ is computed by backpropagation using $\propto |B|W$ operations. If Corner SGD is finitely-approximated using a diagonal matrix $D$ as in Proposition 2, then the number of operations in the linear transformation is $O(MW)$. Accordingly, if $|B| \gg M$ (which should typically be the case in practice), then the computational cost of the linear transformation is negligible compared to the batch gradient estimation, and so the computational overhead of Corner SGD is negligible compared to plain SGD.

**Practical and theoretical acceleration.** Our MNIST experiment in Section F shows that finitely-approximated Corner SGD developed in Section 5 can practically accelerate learning even on realistic problems that are not exactly linear. We note, however, that, in contrast to the ideal infinite-memory Corner SGD of Section 4, this finitely-approximated Corner SGD does not theoretically accelerate the convergence exponent $\zeta$ as $t \to \infty$. (As shown in [29], this is generally impossible for stationary algorithms with finite linear memory.) Nevertheless, we expect that such an acceleration can be achieved with a suitable *non-stationary* approximation. In [29], an acceleration with a factor $\theta$ up to $2 - 1/\nu$ was heuristically derived for a suitable non-stationary memory-1 SGD algorithm.

We remark also that if the model includes nonlinearities, then even the plain SGD in the signal-dominated regime may show a complex picture of convergence rates depending on the strength of the feature learning effects. In particular, [6] consider a particular model where the "rich training" regime is argued to accelerate the "lazy training" exponent $\zeta$ by the factor $\frac{2}{1+\zeta}$. This is different from our factor $\theta_{\max} = \min\left(2, \nu, \frac{2}{\zeta+1/\nu}\right)$ due to a different acceleration mechanism.

# H  The synthetic 1D example

Recall that in Section F we consider the synthetic 1D example in which we fit the target function $y(x) = \mathbf{1}_{[1/4, 3/4]}(x)$ on the segment $[0, 1]$ with a model that in the infinite-size limit has the integral form

$$\widehat{y}(x) = \int_0^1 w(y)(x - y)_+ dy = \mathbf{x}^T \mathbf{w}, \tag{225}$$

where $\mathbf{x}, \mathbf{w}$ are understood as vectors in $L^2([0, 1])$, and $\mathbf{x} \equiv (x - \cdot)_+$. We consider the loss $L(\mathbf{w}) = \mathbb{E}\frac{1}{2}(\mathbf{x}^T \mathbf{w} - y(x))^2$, where $\rho$ is the uniform distribution on $[0, 1]$.

The asymptotic power-law structure of this problem can be derived either from general theory of singular operators and target functions, or from the specific eigendecomposition available in this simple 1D setting.

**The eigenvalues.**  First observe that the operator $\mathbf{H} = \mathbb{E}_{\mathbf{x} \sim \rho}[\mathbf{x}\mathbf{x}^T]$ in our case is the integral operator

$$\mathbf{H}f(x) = \int_0^1 K(x, y)f(y)dy, \quad K(x, y) = \int_0^1 (x - z)_+(y - z)_+ dz. \tag{226}$$

The operator has eigenvalues (see, e.g., Section A.6 of [28]) $\lambda_k = \xi_k^{-4}$, where

$$\xi_k = \frac{\pi}{2} + \pi k + O(e^{-\pi k}), \quad k = 0, 1, \dots \tag{227}$$

Numerically, $\xi_0 \approx 1.875$ so the leading eigenvalue $\lambda_0 \approx 0.0809$.

In particular, the capacity condition (11) holds with $\nu = 4$.

In fact, such a power-law asymptotics is a general property of integral operators with diagonal singularities of a particular order [5]. It is easily checked that the diagonal singularity of operator (226) is of order $\alpha = 3$. In dimension $d$ the exponent $\nu$ has the general form $\nu = 1 + \frac{\alpha}{d}$, which evaluates to 4 in our case $d = 1$.

**The eigencoefficients.**  To establish the source condition (12), we can invoke the general theory that says that for targets that are indicator function of smooth domains we have $\zeta = \frac{1}{d+\alpha} = \frac{1}{4}$ [26]. Alternatively, we can directly find $\zeta$ thanks to the simple structure of the problem.

A short (though not quite rigorous) argument is to observe that the exact minimizer $\mathbf{w}_*$ making the loss $L(\mathbf{w}) = 0$ formally has the distributional form

$$\mathbf{w}_*(x) = \delta'(x - 1/4) - \delta'(x - 3/4) \tag{228}$$

with Dirac delta $\delta(x)$. This vector $\mathbf{w}_*$ has an infinite $L^2([0, 1])$ norm, in agreement with our expectation that $\zeta = \frac{1}{4} < 1$. The eigenfunctions of the problem can be explicitly found (Section A.6 of [28]):

$$\mathbf{e}_k(x) = \cosh(\xi_k x) + \cos(\xi_k x) - \frac{\cosh(\xi_k) + \cos(\xi_k)}{\sinh(\xi_k) + \sin(\xi_k)}(\sinh(\xi_k x) + \sin(\xi_k x)). \tag{229}$$

Then, formally,

$$\mathbf{e}_k^T \mathbf{w}_* = \frac{d\mathbf{e}_k(x)}{dx}\Big|_{x=3/4} - \frac{d\mathbf{e}_k(x)}{dx}\Big|_{x=1/4} \propto \xi_k. \tag{230}$$

It follows that at small $\lambda$, denoting $k_*(\lambda) = \min\{k : \lambda_k < \lambda\}$,

$$\sum_{k:\lambda_k<\lambda} \lambda_k(\mathbf{e}_k^T \mathbf{w}_*)^2 \propto \sum_{k \le k_*(\lambda)} \xi_k^{-2} \propto \sum_{k \le k_*(\lambda)} (1/2 + k)^{-2} \propto k_*^{-1}(\lambda) \propto \lambda^{-1/4}, \tag{231}$$

873    implying again $\zeta = \frac{1}{4}$.

874    A rigorous proof, avoiding Dirac deltas, can be given along the following lines. First note that in the
875    setting of loss function $L(\mathbf{w}) = \frac{1}{2}\mathbb{E}_{\mathbf{x}\sim\rho}(\mathbf{x}^T\mathbf{w} - y(\mathbf{x}))^2$ the vector $\mathbf{q}$ appearing in quadratic form (2)
876    acquires the form $\mathbf{q} = \mathbb{E}_{\mathbf{x}\sim\rho}[y(\mathbf{x})\mathbf{x}]$, which in our example gives

$$\mathbf{q}(x) = \int_{1/4}^{3/4} (y - x)_+ dy. \tag{232}$$

877    We get from the condition $\mathbf{H}\mathbf{w}_* = \mathbf{q}$ that

$$\mathbf{e}_k^T \mathbf{w}_* = -\frac{\mathbf{e}_k^T \mathbf{q}}{\lambda_k}. \tag{233}$$

878    The eigenfunctions can be written as

$$\mathbf{e}_k(x) = \cos(\xi_k x) - \sin(\xi_k x) + e^{-\xi_k x} + (-1)^k e^{-\xi_k(1-x)} + O(e^{-\xi_k}), \tag{234}$$

879    where the last $O(e^{-\xi_k})$ is uniform in $x \in [0,1]$. Performing integration by parts twice with vanishing
880    boundary terms, we find that

$$\mathbf{e}_k^T \mathbf{q} = \int_0^1 \left( \cos(\xi_k x) - \sin(\xi_k x) + e^{-\xi_k x} + (-1)^k e^{-\xi_k(1-x)} \right) \int_{1/4}^{3/4} (y-x)_+ dy dx + O(e^{-\xi_k})$$

$$= -\xi_k^{-1} \int_0^1 \left( \sin(\xi_k x) + \cos(\xi_k x) - e^{-\xi_k x} + (-1)^k e^{-\xi_k(1-x)} \right) \int_{1/4}^{3/4} \mathbf{1}_{y>x} dy dx + O(e^{-\xi_k})$$

$$= \xi_k^{-2} \int_{1/4}^{3/4} (-\cos(\xi_k x) + \sin(\xi_k x)) dx + O(e^{-\xi_k/4}) \tag{235}$$

$$= \xi_k^{-3} \left( -\sin(\pi(\tfrac{1}{2} + k)x) - \cos(\pi(\tfrac{1}{2} + k)x) \right)\Big|_{1/4}^{3/4} + O(e^{-\xi_k/4}) \tag{236}$$

$$\propto \xi_k^{-3}, \tag{237}$$

881    leading to $\mathbf{e}_k^T \mathbf{w}_* \propto \xi_k^{-3}/\lambda_k = \xi_k$, in agreement with Eq. (230).

# I    Extending the proof of Theorem 3 to $\tau_2 \neq 0$

883    In this section we sketch (without much rigor) an argument suggesting that Theorem 3 remains valid
884    under assumption of SE approximation with $\tau_2 \neq 0$ at least if the batch size $|B|$ is large enough.

885    Recall that the assumption $\tau_2 = 0$ was used to write the propagators $U_t, V_t$ in the simple form
886    (8). These representations led to the representations (19), (21) of $U_t, V_t$ in terms of the contour
887    map $\Psi$ that were instrumental in proving Theorem 3. While we are not aware of a similar contour
888    representation at $\tau_2 \neq 0$, we can expand the general $\tau_2 \neq 0$ propagators in terms of the spectral
889    components of the $\tau_2 = 0$ propagators, and in this way reduce the study of the general case to the
890    already analyzed special case.

891    Specifically, let us introduce the notation

$$G_0(t, \lambda) \equiv U^2(t, \lambda) = |(\begin{smallmatrix} 1 & \mathbf{0}^T \end{smallmatrix}) S_\lambda^{t-1} (\begin{smallmatrix} -\alpha \\ \mathbf{c} \end{smallmatrix})|^2. \tag{238}$$

892    Then formula (8) for the propagator $U_t$ can be written as

$$U_t = \frac{\tau_1}{|B|} \sum_{k=1}^\infty \lambda_k^2 G_0(t, \lambda_k). \tag{239}$$

893    In the proof of Theorem 3 it was shown that (see Eqs. (83), (85))

$$G_0(t, \lambda) = U^2(t, \lambda) \approx \lambda^{2/\theta - 2} F_U^2(t\lambda^{1/\theta}). \tag{240}$$

Upon substituting $t\lambda^{1/\theta} = r$ and applying the capacity condition (11), this gave the leading term in $U_t$:

$$U_t \approx \frac{\tau_1}{|B|} \sum_{k=1}^{\infty} \lambda_k^{2/\theta} F_U^2(t\lambda_k^{1/\theta}) \tag{241}$$

$$= \left[ \frac{\tau_1}{|B|} \sum_{k=1}^{\infty} (t\lambda_k^{1/\theta})^2 F_U^2(t\lambda_k^{1/\theta}) \right] t^{-2} \tag{242}$$

$$\approx \left[ \frac{\tau_1}{|B|} \int_{\infty}^{0} r^2 F_U^2(r) d\Lambda^{1/\nu} (t/r)^{\theta/\nu} \right] t^{-2} \tag{243}$$

$$= \left[ \frac{\tau_1}{|B|} \Lambda^{1/\nu} \int_{\infty}^{0} r^2 F_U^2(r) dr^{-\theta/\nu} \right] t^{\theta/\nu - 2}. \tag{244}$$

Now, if the SE approximation holds with $\tau_2 \neq 0$, then the propagator formulas (8) are no longer valid. Instead (see [29]), the propagators can be written with the help of the linear transition operators $A_\lambda$ acting on $(M+1) \times (M+1)$ matrices $Z$:

$$A_\lambda Z = S_\lambda Z S_\lambda^T - \frac{\tau_2}{|B|} \lambda^2 \left( \begin{smallmatrix} -\alpha \\ \mathbf{c} \end{smallmatrix} \right) \left( \begin{smallmatrix} 1 \\ \mathbf{0} \end{smallmatrix} \right)^T Z \left( \begin{smallmatrix} 1 \\ \mathbf{0} \end{smallmatrix} \right) \left( \begin{smallmatrix} -\alpha \\ \mathbf{c} \end{smallmatrix} \right)^T. \tag{245}$$

In particular, Eqs. (238), (239) get replaced by

$$U_t = \frac{\tau_1}{|B|} \sum_{k=1}^{\infty} \lambda_k^2 G(t, \lambda_k), \tag{246}$$

$$G(t, \lambda) = \mathrm{Tr}[ \left( \begin{smallmatrix} 1 \\ \mathbf{0} \end{smallmatrix} \right) \left( \begin{smallmatrix} 1 \\ \mathbf{0} \end{smallmatrix} \right)^T A_\lambda^{t-1} [ \left( \begin{smallmatrix} -\alpha \\ \mathbf{c} \end{smallmatrix} \right) \left( \begin{smallmatrix} -\alpha \\ \mathbf{c} \end{smallmatrix} \right)^T ]]. \tag{247}$$

Note that Eq. (238) is a special case of Eq. (247) resulting at $\tau_2 = 0$ thanks to the simple factorized structure of the transformation $A_\lambda$ with vanishing second term.

Let us now write the binomial expansion of $G(t, \lambda)$ by choosing one of the two terms on the r.h.s. of Eq. (245) in each of the $t-1$ iterates of $A_\lambda$ in Eq. (247). The key observation here is that each term in this binomial expansion can be written as a product of the $\tau_2 = 0$ factors $G_0$ with a suitable coefficient:

$$G(t, \lambda) = G_0(t, \lambda) + \sum_{m=1}^{t-1} \left( \frac{-\tau_2 \lambda^2}{|B|} \right)^m \times \tag{248}$$

$$\times \sum_{0 < t_1 < \ldots < t_m < t} G_0(t - t_m, \lambda) G_0(t_m - t_{m-1}, \lambda) \cdots G_0(t_2 - t_1, \lambda) G_0(t_1, \lambda). \tag{249}$$

Here, $0 < t_1 < \ldots < t_m < t$ are the iterations at which the second term in Eq. (245) was chosen.

We can now apply again approximation (240) for $G_0$ in terms of $F_U$, and approximate summation by integration:

$$G(t, \lambda) \approx \lambda^{2/\theta - 2} \left[ F_U^2(t\lambda^{1/\theta}) + \sum_{m=1}^{\infty} \left( \frac{-\tau_2 \lambda^{1/\theta}}{|B|} \right)^m (F_U^2)^{*(m+1)}(t\lambda^{1/\theta}) \right], \tag{250}$$

where $(F_U^2)^{*(m+1)}$ is the $(m+1)$-fold self-convolution of $F_U^2$:

$$(F_U^2)^{*(m+1)}(r) = \int \cdots \int_{0 < r_1 < \ldots r_m < r} F_U^2(r - r_m) F_U^2(r_m - r_{m-1}) \cdots F_U^2(r_1) dr_1 \cdots dr_m. \tag{251}$$

The factor $\lambda^{1/\theta}$ in (250) results from the respective factor $\lambda^2$ in Eq. (248), the factor $\lambda^{2/\theta - 2}$ in Eq. (240), and the integration element scaling factor $\lambda^{-1/\theta}$ due to the substitution $r_n = t_n \lambda^{1/\theta}$.

The leading term in expansion (250) corresponds to the case $\tau_2 = 0$. Consider the next term, $m = 1$. The respective contribution to $U_t$ is

$$U_t^{(1)} \equiv -\frac{\tau_1 \tau_2}{|B|^2} \sum_{k=1}^{\infty} \lambda_k^{3/\theta} (F_U^2)^{*2}(t\lambda_k^{1/\theta}). \tag{252}$$

This expression can be analyzed similarly to the leading term in Eq. (241), giving

$$U_t^{(1)} \approx -\left[\frac{\tau_1\tau_2}{|B|^2}\Lambda^{1/\nu}\int_\infty^0 r^3(F_U^2)^{*2}(r)dr^{-\theta/\nu}\right]t^{\theta/\nu-3}. \tag{253}$$

Note the faster decay $t^{\theta/\nu-3}$ compared to $t^{\theta/\nu-2}$ in the leading term. This difference results from the different exponent $3/\theta$ on $\lambda_k$. It also leads to the factor $r^3$ rather than $r^2$ in the integral.

The coefficient in brackets in Eq. (253) is finite unless the integral diverges. To see the convergence, write

$$\int_\infty^0 r^3(F_U^2)^{*2}(r)dr^{-\theta/\nu} = \frac{\theta}{\nu}\int_0^\infty r^{2-\theta/\nu}(F_U^2)^{*2}(r)dr \tag{254}$$

and use the inequality $r^{2-\theta/\nu} \le (2(r-r_1))^{2-\theta/\nu} + (2r_1)^{2-\theta/\nu}$ valid since $2-\theta/\nu > 0$:

$$\int_0^\infty r^{2-\theta/\nu}(F_U^2)^{*2}(r)dr \tag{255}$$

$$\le \int\int_{0<r_1<r<\infty}[(2(r-r_1))^{2-\theta/\nu} + (2r_1)^{2-\theta/\nu}]F_U^2(r-r_1)F_U^2(r_1)dr_1dr \tag{256}$$

$$= 2^{3-\theta/\nu}\left(\int_0^\infty r^{2-\theta/\nu}F_U^2(r)dr\right)\left(\int_0^\infty F_U^2(r)dr\right) < \infty, \tag{257}$$

since $F_U(r) \propto r^{-\theta-1}$ as $r \to \infty$ by Lemma 1.

Next terms in expansion (250) can be analyzed similarly, but we encounter the difficulty that, due to the associated factor $\lambda^{m/\theta}$ in Eq. (250), they will contain the integrals $\int_\infty^0 r^{2+m}(F_U^2)^{*(m+1)}(r)dr^{-\theta/\nu}$ that diverge for sufficiently large $m$. For this reason, it is convenient to upper bound

$$\lambda^{m/\theta} \le \lambda_{\max}^{(m-1)/\theta}\lambda^{1/\theta}. \tag{258}$$

Then the contribution $U_t^{(m)}$ to $U_t$ from the term $m$ can be upper bounded by

$$|U_t^{(m)}| \lesssim \left[\frac{\tau_1|\tau_2|^m\lambda_{\max}^{(m-1)/\theta}}{|B|^{m+1}}\Lambda^{1/\nu}\int_\infty^0 r^3(F_U^2)^{*(m+1)}(r)dr^{-\theta/\nu}\right]t^{\theta/\nu-3}. \tag{259}$$

Using the inequality $r^{2-\theta/\nu} \le ((m+1)(r-r_m))^{2-\theta/\nu} + \ldots + ((m+1)r_1)^{2-\theta/\nu}$, the integral can be bounded as

$$\int_\infty^0 r^3(F_U^2)^{*(m+1)}(r)dr^{-\theta/\nu} \le \frac{\theta}{\nu}(m+1)^{3-\theta/\nu}\left(\int_0^\infty r^{2-\theta/\nu}F_U^2(r)dr\right)\left(\int_0^\infty F_U^2(r)dr\right)^m < \infty. \tag{260}$$

Summarizing, the contribution of all the terms in $U_t$ other than the leading term $U_t^{(0)}$ can be upper bounded by

$$|U_t - U_t^{(0)}| \lesssim Ct^{\theta/\nu-3}, \tag{261}$$

with the constant

$$C = \frac{\tau_1\theta\Lambda^{1/\nu}}{\nu}\left(\int_0^\infty r^{2-\theta/\nu}F_U^2(r)dr\right)\sum_{m=1}^\infty\frac{|\tau_2|^m\lambda_{\max}^{(m-1)/\theta}}{|B|^{m+1}}(m+1)^{3-\theta/\nu}\left(\int_0^\infty F_U^2(r)dr\right)^m. \tag{262}$$

If

$$|B| > |\tau_2|\lambda_{\max}^{1/\theta}\int_0^\infty F_U^2(r)dr, \tag{263}$$

then series (262) converges, and so $|U_t - U_t^{(0)}| = o(U_t^{(0)})$, as claimed.

The case of the propagators $V_t$ can be treated similarly. Starting from $\tau_2 = 0$, denote

$$H_0(t,\lambda) = V^2(t,\lambda) = |(\begin{smallmatrix}1 & \mathbf{o}^T\end{smallmatrix})S_\lambda^{t-1}(\begin{smallmatrix}1\\\mathbf{0}\end{smallmatrix})|^2, \tag{264}$$

then by Eqs. (153), (155) $H_0(t,\lambda) \approx F_V^2(t\lambda^{1/\theta})$ and

$$V_t = \sum_{k=1}^\infty\lambda_k(\mathbf{e}_k^T\mathbf{w}_*)^2 H_0(t,\lambda) \approx \sum_k\lambda_k(\mathbf{e}_k^T\mathbf{w}_*)^2 F_V^2(t\lambda_k^{1/\theta}). \tag{265}$$

The counterpart of $H_0$ for general $\tau_2$ is

$$H(t, \lambda) = \text{Tr}[\left(\begin{smallmatrix}1\\\mathbf{0}\end{smallmatrix}\right)\left(\begin{smallmatrix}1\\\mathbf{0}\end{smallmatrix}\right)^T A_\lambda^{t-1}[\left(\begin{smallmatrix}1\\\mathbf{0}\end{smallmatrix}\right)\left(\begin{smallmatrix}1\\\mathbf{0}\end{smallmatrix}\right)^T]]. \tag{266}$$

Expansion (248) gets replaced by

$$H(t, \lambda) = H_0(t, \lambda) + \sum_{m=1}^{t-1} \left(\frac{-\tau_2\lambda^2}{|B|}\right)^m \times \tag{267}$$

$$\times \sum_{0<t_1<\ldots<t_m<t} G_0(t - t_m, \lambda)G_0(t_m - t_{m-1}, \lambda)\cdots G_0(t_2 - t_1, \lambda)H_0(t_1, \lambda) \tag{268}$$

and expansion (250) gets replaced by

$$H(t, \lambda) \approx F_V^2(t\lambda^{1/\theta}) + \sum_{m=1}^{\infty} \left(\frac{-\tau_2\lambda^{1/\theta}}{|B|}\right)^m ((F_U^2)^{*m} * F_V^2)(t\lambda^{1/\theta}). \tag{269}$$

The factor $\lambda^{m/\theta}$ can again be used to extract an extra negative power of $t$ in the asymptotic bounds. To avoid divergence of the integrals, we can use a bound

$$\lambda^{m/\theta} \leq \lambda_{\max}^{(m-\epsilon)/\theta}\lambda^{\epsilon/\theta} \tag{270}$$

with some sufficiently small $\epsilon > 0$. Arguing as before, we then find that for $|B|$ large enough the contribution of all the terms $m \geq 1$ is $O(t^{-\theta\zeta-\epsilon})$, i.e. asymptotically negligible compared to the leading term $\propto t^{-\theta\zeta}$.

