# OpenReview forum: "Corner Gradient Descent"
_NeurIPS.cc/2025/Conference — Submitted to NeurIPS 2025_

### Official Review · Reviewer_rYcZ · 2025-07-01

**Clarity:** 3
**Significance:** 2
**Originality:** 4
**Rating:** 4
**Confidence:** 3

**Summary:**

This paper considers convergence acceleration in stochastic gradient descent (SGD) under infinite-dimensional quadratic problems with power-law spectral decay. The authors provide a geometric picture identifying SGD algorithms with contours in the complex plane. They now propose corner algorithms, which are characterized by contours with a corner angle of a particular size and yield accelerated convergence rates. They then rigorously derive the sharp form of the attainable acceleration, which is an exact balance between increased shocks from noise at a higher speed of convergence. Furthermore, in the experiments, it is shown that ideal corner algorithms of infinite memory can be well approximated in practice by finite-memory implementations.

**Questions:**

see weakness

**Ethical Concerns:**

["NO or VERY MINOR ethics concerns only"]

**Final Justification:**

most of the concerns are addressed.

**Limitations:**

see weakness

**Paper Formatting Concerns:**

see weakness

**Quality:**

3

**Strengths And Weaknesses:**

Strengths: 1. The paper introduces a different perspective with generalized SGD from the geometric view of complex-plane contours, and it further provides a nuanced acute insight during the designing of these algorithms.
2. It rigorously proves that, under special spectral conditions, corner SGD algorithms can accelerate the convergence.

Weaknesses: 1. The analysis is constrained within the realm of quadratic losses and spectral power-law conditions, which might restrict its applicability to general settings encountered in deep learning.
2. Although mathematically elegant, the infinite-dimensional setting might be considered too idealistic to be truly relevant for the majority of real-world problems.

I think this paper would be more suitable for journals like MP or SIAM series.

---

> ### Author Rebuttal · Authors · 2025-07-30
>
> We thank the reviewer for the review. However, we believe that this review with its score "3: Borderline reject" misrepresents the actual balance of contributions of our work. Our paper includes multiple entirely new results:
> - the algorithm-contour correspondence,
> - the concept of a corner algorithm,
> - the first proof of power-law acceleration,
> - the phase diagram with three phases,
> - an efficient practical approximation of corner algorithms.
>
> All these are new rigorous results. Most of what the reviewer calls "weaknesses" are natural, reasonable, and essential elements of our theoretical setting. Any theory relies on some assumptions and has an applicability range. We believe that our assumptions and applicability range are reasonable and appropriate.
>
> > 1. The analysis is constrained within the realm of quadratic losses and spectral power-law conditions, which might restrict its applicability to general settings encountered in deep learning.
>
> 1. The problem we are addressing is difficult and open even in the quadratic case. It is odd to criticize the work for not handling a much more challenging setting.
>
> 2. Already in the quadratic setting, our solution is novel and complex. The answers we give in this setting - for example, the phase diagram with three phases - are entirely new, varied, rigorous and informative.
>
> 3. The power law spectral condition we use is the most standard theoretical assumption quantitatively characterizing ill-conditioned problems. It is used in numerous works, in particular those we cite.
>
> 4. We never claim in our paper that our algorithm outperforms algorithms for general deep learning. We do claim, prove, and experimentally demonstrate that it accelerates the power law in batch-wise gradient-based learning in ill-conditioned linear regression, which no other algorithm has been shown to achieve.
>
>
>
> > 2. Although mathematically elegant, the infinite-dimensional setting might be considered too idealistic to be truly relevant for the majority of real-world problems.
>
> 1. The infinite-dimensional setting naturally fits a theoretical study of training acceleration in ill-conditioned models at indefinitely large iterations.
>
> 2. The fact that the theoretical dimensionality is infinite does not make the work practically irrelevant. Infinite abstractions are commonly used in ML research. Even the "$O(..)$" bounds ubiquitous in ML literature mathematically make sense only for infinitely many observations, which never happens in real-world problems.
>
> 3. While our ideal corner algorithm requires infinite memory, we directily address practical applicability by deriving a finite-memory approximation in section 5. We also theoretically explain there why such an approximation can be performed very efficiently thanks to the special structure of the argorithm.
>
> 4. An implementation of our algorithm, along with demonstrations, is provided in the jupyter notebook in the Supplementary Material. It can be easily seen that our algorithm is not much harder to use in practice than, for example, basic SGD with momentum.
>
> 5. While our theorems are stated under particular assumptions, this does not mean that our algorithm can practically be used only if these assumptions hold. As we discuss in response to reviewer CvZn, it is usable even in strongly nonquadratic settings such as CIFAR10/Resnet. In deep learning, there is a big gap between practical performance of algorithms and rigorously established guarantees. In particular, momentum-type constructions are well-known to accelerate convergence, but we are not aware of any proof of that for a real world neural network not involving substantial mathematical approximations and simplifications.
>
>
> In conclusion, we don't see this review pointing out any actual sigificant drawbacks in our work that would offset our contributions. "Weaknessess" of the kind indicated in the review (not considering a more ambitious problem, using mathematical approximations and simplifed models) can be found in virtually any theoretical work presented at NeurIPS. We don't think that the score "3: Borderline reject" is properly justified by this review.

---

> > ### Comment · Reviewer_rYcZ · 2025-08-06
> > **thanks for the response**
> >
> > thanks for the response, I raised the score to 4.

---

### Official Review · Reviewer_CvZn · 2025-07-02

**Clarity:** 3
**Significance:** 3
**Originality:** 3
**Rating:** 4
**Confidence:** 3

**Summary:**

The paper introduces **Corner Gradient Descent (Corner SGD)**, a family of stationary stochastic gradient algorithms that achieve *accelerated* convergence on ill-conditioned quadratic objectives with power-law spectra.

**Questions:**

1. **SE assumption & $τ_2\neq0$.** Can you provide numerical evidence (or a theorem) that Theorem 3’s rates persist when $τ_2>0$ for realistic data distributions? Clarifying this would raise confidence in practical relevance.
2. **Memory–batch trade-off.** Your complexity discussion assumes $|B|\gg M$. How sensitive is performance when $M$ approaches the mini-batch size, e.g., micro-batch settings on GPUs?
3. **Non-quadratic losses.** Experiments show gains on MNIST in the NTK regime. Do corner algorithms still help once feature learning dominates (far from linearisation)? Could you test on CIFAR-10 with a deeper model?
4. **Adaptive θ.** In practice θ\_max depends on unknown spectral exponents. Do you foresee an online procedure (e.g. based on loss curvature estimates) to tune θ automatically?
5. **Stability margins.** Figure 5 shows instability for θ = 1.8 with small batches. Could you derive an explicit batch-size lower bound ensuring $U_\Sigma<1$?

**Ethical Concerns:**

["NO or VERY MINOR ethics concerns only"]

**Quality:**

3

**Strengths And Weaknesses:**

| Dimension        | Strengths                                                                                                                                                                        | Weaknesses                                                                                                                                                                                                                    |
| ---------------- | -------------------------------------------------------------------------------------------------------------------------------------------------------------------------------- | ----------------------------------------------------------------------------------------------------------------------------------------------------------------------------------------------------------------------------- |
| **Quality**      | • Rigorous spectral analysis with clear stability conditions.• Tight phase diagram balancing acceleration vs. noise.• Implementation recipe with complexity discussion.  | • Core theory assumes Spectrally-Expressible (SE) approximation with $τ_2=0$; extension to $τ_2\neq0$ is only sketched. • Infinite-memory algorithm is not implementable; finite-memory variant loses proven asymptotics. |
| **Clarity**      | • Paper well structured: geometry (§3) → theory (§4) → approximation (§5) → experiments.• Helpful figures and derivations in appendices.                                     | • Dense mathematics; main intuition occasionally buried in technical lemmas.• Notation heavy—e.g., simultaneous use of $P/Q$, $Ψ$, and propagators may confuse readers.                                                   |
| **Significance** | • Advances long-standing question of SGD acceleration under noise.• Provides unifying geometric viewpoint that could influence future optimiser design.                      | • Applicability limited to power-law spectra and near-quadratic regimes; real-world deep nets rarely satisfy these assumptions.• Empirical validation restricted to MNIST and a toy 1-D kernel; no large-scale tasks.     |
| **Originality**  | • Novel contour representation and “corner” construction.• Bridges complex analysis with optimisation theory in an original way.                                             | • Connection to existing non-stationary heavy-ball schedules is mostly qualitative.                                                                                                                                           |

---

> ### Author Rebuttal · Authors · 2025-07-30
>
> Thank you for the careful reading of our work and very interesting questions.
>
> 1. SE assumption and $\tau_2\ne 0$.
>
> It can be seen from the formulation of SE approximation (6) that the parameter $\tau_2$ represents a subleading term and should not affect the results as much as parameter $\tau_1$. Namely, suppose that the matrix $\mathbf C$ is diagonal in the eigenbases of $H$. Then in the $\lambda$-eigenspace of $H$, the term $\tau_2 HCH$ multiplies the eigenvalue of $C$ by $\lambda^2$, while the term $\tau_1 Tr[HC]H$ replaces it by $c\lambda$, with $c=Tr[CH]$. At large iterations, the main challenge is to suppress the spectral components with small $\lambda$. Accordingly, the term  $\tau_2 HCH$ makes only a small, quadratic and localized contribution to the evolution. In contrast, the effect of the term $\tau_1 Tr[HC]H$ is linear in $\lambda$ and depends on the components of $C$ in all eigenspaces. Thus, at large iterations we expect the $\tau_1$ term to dominate. This is confirmed by the fact that in the general formulas (246), (247) for the propagators $U_t$ with an arbitrary $\tau_2$, the parameter $\tau_1$ determines the overall coefficient, while $\tau_2$ only determines the second order correction.
>
> However, despite these heuristics, handling the $\tau_2$ term fully rigorously does present a difficulty. It seems that the $\tau_2=0$ model is a natural "solvable" model. Models with $\tau_2\ne 0$ should by qualitatively similar, but they are not solvable in a similar nice way.
>
> 2. Memory–batch trade-off.
>
> As we discuss in section G, computational complexity grows linearly with $M$ due to additions and multiplications by constants of the auxiliary vectors and the gradient. If the batch size is comparable to $M$ (in our experiments, for example, we take $M=5$, but a smaller number may be sufficient for some tasks), then it becomes important to compare the gradient computation for a single data point with the mentioned additions and multiplications for a single auxiliary vector. In terms of simple count of arithmetic operations, the gradient requires more operations, so even in the case of $|B|$ comparable to $M$ the computational overhead may be not very significant in terms of operation count. However, the reality may be, of course, more complicated due to hardware implementation issues, optimized parallelization, etc.
>
>
> 3. Non-quadratic losses.
>
> We expect that corner algorithms can help at least if feature learning is not too fast and the models on the learning trajectory have quadratic approximations belonging to the accelerated region of our spectral phase diagram. Unfortunately, in the case of CIFAR 10 with Resnets the quadratic approximation is very close to the boundary of accelerated region - there is actually a respective point in figure 1 (left). The respective $\nu$ approximately equals 1.1, implying that we cannot expect acceleration beyond $\theta\approx 1.1$. In practice, we do observe some improvement over SGD in this case, but only very slight (e.g., the loss reached after 10 epochs with SGD can be reached after 9 epochs with the corner algorithm).
>
> 4.  Adaptive $\theta.$
>
> This is an interesting possibility that we haven't explored. There is some analogy between $\theta$ in the corner algorithm and the momentum parameter $\beta$ in SGD with momentum, so the practice of tuning $\theta$ might be expected to resemble the existing practice of tuning $\beta$.
>
> 5. Stability margins.
>
> It's an interesting question. We can roughly estimate the critical batch size for $U_\\Sigma < 1$ from the integration-summation formula (23). Estimation of the intergral gives something like $|B|_{crit}\\propto \\frac{\\tau_1}{(2\\pi)^2(2-\\theta)}\\sum_k \\lambda_k^{1/\\theta}$.
>
> Assuming $\\lambda_k \\sim k^{-\\nu},$ if $\\theta < \\nu$, then the critical batch size $|B|_{crit}\\propto\\frac{\\tau_1}{(2\\pi)^2(2-\\theta)(\\nu/\\theta-1)}$.
> As expected, the critical batch size diverges as $\\theta$ approaches 2 or $\\nu$.
>
> If $\\theta > \\nu$ and the dimension of the space is finite $N$, then $|B|_{crit}\\propto\\frac{\\tau_1}{(2\\pi)^2(2-\\theta)(1-\\nu/\\theta)}N^{1-\\nu/\\theta}$.
>
> In the MNIST case with $\\theta=1.8,\\nu\\approx 1.3, N=10000$, this gives $|B|_{crit} \\sim 10$. This is lower than $|B|=100$ for which we observe an instability in figure 6. However, the instability there is present only at very late iterations, and the spectral properties of the problem there may be different from the initial estimate $\\nu=1.3$. Also, the experiment is performed not with the ideal corner algorithm, but with its approximation with memory $M=5$.

---

### Official Review · Reviewer_6Dvj · 2025-07-03

**Clarity:** 2
**Significance:** 3
**Originality:** 4
**Rating:** 4
**Confidence:** 3

**Summary:**

This paper studies SGD algorithms on infinite-dimensional quadratic problems with power law spectral conditions. The authors propose to use stationary SGD algorithms with an arbitrary-sized linear memory. By viewing it as a contour, one can identify a corner with external angle that can be used to accelerate the SGD algorithm to $O(t^{-2\zeta})$ in one phase of the algorithm dynamics. This result address the limitation of Heavy Ball method in mini-batch SGD.

**Questions:**

1. Can the authors add a section to discuss the limitations of the proposed algorithm and its finite-memory approximation?

**Ethical Concerns:**

["NO or VERY MINOR ethics concerns only"]

**Final Justification:**

I have read other reviews and rebuttals. I decided to maintain my score.

**Limitations:**

n.a.

**Paper Formatting Concerns:**

n.a.

**Quality:**

4

**Strengths And Weaknesses:**

Strengths:
1. The proposed method introduces a new tool or viewpoint to study generalized SGD method.
2. The achieved acceleration is novel in the literature.

Weakness:
1. The achieved acceleration is only identified in one of the three phases.
2. There lacks a detailed introduction to the considered problem and related works.
3. The infinite memory condition needs to be justified.

---

> ### Author Rebuttal · Authors · 2025-07-30
>
> We thank the reviewer for the review.
>
> > Can the authors add a section to discuss the limitations of the proposed algorithm and its finite-memory approximation?
>
> In fact we have section G dealing primarily with these issues. Due to the page limit, we had to move this section to the appendix. We see our main theoretical limitation in the complete rigorous proof being given under the relatively strict assumption of spectrally-expressible approximation with parameter $\tau_2=0$ (which corresponds to a "solvable" model in some sense). Practically, our main limitation is a memory and computation overhead compared to plain SGD. We discuss these limitations in detail in Section G, and discuss how they can be mitigated. If the final version allows an extra page, we will move this section back to the main text.

---

### Official Review · Reviewer_WeqW · 2025-07-06

**Clarity:** 3
**Significance:** 3
**Originality:** 3
**Rating:** 5
**Confidence:** 1

**Summary:**

This paper introduces a new analysis of (S)GD with complex analysis and a new optimization algorithm.

**Questions:**

N/A

**Ethical Concerns:**

["NO or VERY MINOR ethics concerns only"]

**Quality:**

3

**Strengths And Weaknesses:**

This paper is beyond my understanding of optimization, algebra and complex analysis, even though I tried really hard. I personlly think it is a very solid work comparing to other papers, I recommend someone else to review it.

---

### Decision · Program_Chairs · 2025-09-17

**Decision:**

Reject

**Comment:**

The main focus of the paper is on the design and analysis of accelerated SGD algorithms for infinite dimensional quadratic problems with power law spectral decay. For this class of problems, the paper shows that acceleration is possible even in the presence of noise in an idealized setting with infinite memory. The paper also gives a finite memory approximation of the idealized algorithm but these approximations do not retain the accelerated rate. The reviewers appreciated the novel geometric perspective introduced in this work and the theoretical contribution. The reviewers raised concerns regarding the restrictive problem setting considered and the idealized approach with infinite memory, which significantly restricts the applicability of the proposed algorithms. Overall, although the paper provides a valuable perspective and approach for the design of accelerated SGD algorithms, the setting considered is quite restrictive and its potential applications may be limited.